# A titin missense variant drives atrial electrical remodeling and is associated with atrial fibrillation

**Mahmud Arif Pavel[1]\*[†], Hanna Chen[1†], Michael Hill[1†], Arvind Sridhar[1], Miles Barney[1], Jaime DeSantiago[1], Abhinaya Baskaran[1], Asia Owais[1], Shashank Sandu[1], Faisal A Darbar[1], Aylin Ornelas Loredo[1], Bahaa Al-Azzam[1], Brandon Chalazan[2,3], Jalees Rehman[1,4], Dawood Darbar[1,5,6]\***

[1]Division of Cardiology, Department of Medicine, University of Illinois at Chicago, Chicago, United States; [2]Division of Genetics, Genomics, and Metabolism, Department of Pediatrics, Lurie Children's Hospital of Chicago, Chicago, United States; [3]Department of Pharmacology, Northwestern University, Chicago, United States; [4]Department of Biochemistry and Molecular Genetics, University of Illinois at Chicago, Chicago, United States; [5]Department of Pharmacology, University of Illinois at Chicago, Chicago, United States; [6]Jesse Brown Veterans Administration Medical Center, Chicago, United States

**\*For correspondence:**
gmmahmud@uic.edu (MAP);
darbar@uic.edu (DD)

[†]These authors contributed equally to this work.

## eLife Assessment

The study presents **important** findings regarding the incidence and clinical impact of a mutation in a cardiac muscle protein and its association with the development of atrial fibrillation. The authors provide **convincing** evidence of electrophysiological disturbances in cells with this mutation and of its association with atrial fibrillation, which would be of interest to cardiologists. Evidence supporting the conclusion that this mutation causes atrial fibrillation would benefit from more rigorous electrophysiologic approaches.

**Abstract** Rare and common genetic variants contribute to the risk of atrial fibrillation (AF). Although ion channels were among the first AF candidate genes identified, rare loss-of-function variants in structural genes, such as *TTN*, have also been implicated in AF pathogenesis, partly through the development of atrial myopathy; however, the underlying mechanisms are poorly understood. While *TTN* truncating variants (*TTN*tvs) have been causally linked to arrhythmia and cardiomyopathy syndromes, the role of missense variants (mvs) remains unclear. We show that rare *TTNmvs* are associated with worse clinical outcomes in a single-center ethnic minority clinical cohort and uncover a pathogenic mechanism by which the T32756I variant drives AF. Modeling the *TTN*-T32756I variant using human induced pluripotent stem cell-derived atrial cardiomyocytes (iPSC-aCMs) revealed that the mutant cells display aberrant contractility, increased activity of a cardiac potassium channel (KCNQ1, Kv7.1), and dysregulated calcium homeostasis without compromising the sarcomeric integrity of the atrial cardiomyocytes. We also show that a titin-binding protein, the Four-and-a-Half Lim domains 2 (FHL2), has increased binding with KCNQ1 and its modulatory subunit KCNE1 in the *TTN*-T32756I-iPSC-aCMs, enhancing the slow delayed rectifier potassium current ($I_{ks}$). Suppression of FHL2 in mutant iPSC-aCMs normalized the $I_{ks}$, supporting FHL2 as an $I_{ks}$ modulator. Our findings demonstrate that a single amino acid substitution in titin not only impairs its function but also remodels ion channels, contributing to AF. These findings underscore the importance of high-throughput screening to assess the pathogenicity of *TTN*mvs and establish a mechanistic connection

between titin, potassium ion channels, and sarcomeric proteins, which may represent a novel therapeutic target.

## Introduction

Atrial fibrillation (AF), the most prevalent cardiac arrhythmia, affects more than 60 million people worldwide and is associated with increased risk for stroke, heart failure, and dementia, justifying it as a major public healthcare burden (*Brundel et al., 2022*; *Elliott et al., 2023*). AF is characterized by irregular and often abnormally fast heart rates (*Schotten et al., 2011*; *Markides and Schilling, 2003*). Over the last decade, tremendous progress has been made in understanding the genetic architecture of AF (*Lee et al., 2023*; *Feghaly et al., 2018*; *Sagris et al., 2022*). Genome-wide association studies have identified over 140 common loci associated with AF, while family-based studies have implicated rare variants primarily encoding ion channels (*Roselli et al., 2020*; *Roselli et al., 2018*). Although AF has been traditionally classified as a 'channelopathy', variants in myocardial sarcomeric proteins such as titin have been increasingly associated with familial or early-onset AF (*Choi et al., 2018*; *Chalazan et al., 2021*).

The *TTN* gene encodes for a massive myofilament (~4200 kDa), titin, which stretches along the Z-disk (N-terminus) to the M-band (C-terminus) region of the sarcomere (*Labeit and Kolmerer, 1995*; *Granzier and Labeit, 2004*). Titin serves as a molecular scaffold for other muscle proteins, participates in downstream signaling, and provides passive tension in cardiac muscle (*LeWinter et al., 2007*; *Linke and Hamdani, 2014*). Due to its large size, the rate of genetic variation in *TTN* is high, including both truncating (*TTN*tvs) and missense variants (*TTN*mvs) (*Schafer et al., 2017*; *Begay et al., 2015*). The prevalence of rare *TTN*tvs and *TTN*mvs in the general population is 2% and 5.7%, respectively (*Vissing et al., 2021*; *Akinrinade et al., 2019*; *Weston et al., 2024*). While *TTN*tvs are the most common genetic cause (10–20%) of dilated cardiomyopathy (DCM), *TTN*mvs are often disregarded in clinical practice and are considered benign in DCM-specific variant interpretation frameworks based on the American College of Medical Genetics (ACMG) recommendations (*Martínez-Barrios et al., 2022*; *Richards et al., 2015*). Yet, *TTN*mvs may have the potential to confer disease, as a recent study in two families revealed that *TTN*mvs in a conserved cysteine of *TTN* can cause DCM (*Domínguez et al., 2023*). Another study demonstrated segregation of a novel *TTN*mv in five individuals with atrioventricular block in a Chinese family, suggesting that *TTN*mvs may also be implicated in arrhythmia syndromes (*Liu et al., 2020*). To date, a relationship between rare *TTN*mvs and AF has not been explored at either clinical or mechanistic levels. This has potentially heightened importance across racial and ethnic groups, in whom the likelihood of detecting variants of uncertain significance (which are predominantly *TTN*mvs) is higher than in individuals of European descent (*Chen et al., 2023*).

Despite the clinical importance, the pharmacological therapy of AF is limited in part because of the incomplete understanding of the myocardial substrate for AF, as human atrial tissue is rarely available, and the limitations of existing in vitro and in vivo models (*Ly et al., 2021*). However, human-induced pluripotent stem cell-derived atrial cardiomyocytes (iPSC-CMs) not only possess the complex array of cardiac ion channels that make up the atrial action potential (AP) but also hold great promise for modeling AF (*Ly et al., 2022*; *Nicholson et al., 2022*). Modeling patient-specific mutations associated with familial AF using mature human iPSC-CMs offers a powerful, naturally integrated system with distinct advantages over animal models and heterologous expression systems (*Hong et al., 2021*). Recent reports using human iPSC-CM models have elucidated the pathophysiological mechanisms of DCM (*Poetsch and Guan, 2020*), hypertrophic cardiomyopathy (*Li et al., 2018*), Brugada syndrome (*Nijak et al., 2021*), long QT syndrome (*Moretti et al., 2010*), and AF (*Ly et al., 2022*; *Hong et al., 2021*; *Benzoni et al., 2020*). Human iPSC-CMs also have the potential to uncover the molecular mechanisms of AF, as they can be easily modified to contain the precise genetic background of the individual patient (*Simons et al., 2023*; *Brown et al., 2024*; *Jiang et al., 2024*).

To examine the potential role of *TTN*mvs in AF, we examined the prevalence of rare *TTN*mvs in a single-center cohort of African American and Hispanic/Latinx individuals with AF who underwent whole exome sequencing, compared clinical characteristics between *TTN*mv carriers and non-carriers, and evaluated whether *TTN*mvs were associated with increased risk of AF or heart failure (HF)-related hospitalizations. To explore whether the *TTN*mvs may be mechanistically linked to the development of AF, we introduced a rare *TTN*mv (*TTN*-T32756I) into human iPSC-aCMs using clustered regularly

interspaced short palindromic repeats-associated 9 (CRISPR-Cas9). We performed electrophysiological (EP), pharmacological, and mechanistic analyses to elucidate the mechanisms underlying *TTN*mv-induced AF. Our study demonstrated that *TTN*-T32756I iPSC-aCMs exhibited a striking AF-like EP phenotype in vitro. Transcriptomic analyses revealed that this *TTN*mv enhances the activity of the Four-and-a-half LIM domain protein 2 (FHL2), which, in turn, modulates the slow delayed rectifier potassium current ($I_{Ks}$), thereby promoting AF susceptibility.

## Results

### *TTN*mvs are associated with increased hospitalization risk in a multiethnic cohort with AF

#### Clinical characteristics

A total of 131 subjects were included (mean [SD] age at AF diagnosis 63.5 [13.8] years, 70 [53.4%] male, 38 [29%] Hispanic/Latinx, 93 [71%] non-Hispanic Black; *Table 1*). We identified 138 *TTN*mvs, most commonly in the *TTN* A-band (108 [78.8]%), followed by the I-band (20 [14.6%]), M-band (2 [1.5%]), near the Z-disk (6 [4.4%]), and in the Z-disk (2 [1.5%]) (*Figure 1A*, *Supplementary file 1*). Based on the REVEL in silico score, 52 (37.7%) *TTN*mvs variants were predicted to be deleterious. A total of 77 (58.8%) subjects carried a *TTN*mv, with 43 (32.8%) carrying a predicted deleterious variant. Carriers of a *TTN*mv had a higher QTc interval on electrocardiogram (ECG) closest to AF diagnosis (mean [SD] 466.5 [42.3] vs. 449.6 [37.7] ms, p=0.027); there were otherwise no other significant clinical or demographic differences between variant carriers and non-carriers. When stratifying by predicted deleterious *TTN*mv only (*Supplementary file 2*), a higher proportion of variant carriers were non-Hispanic Black (36 [83.7%] vs. 57 [64.8%], p=0.026). On index ECG, carriers additionally had higher ventricular rate (mean [SD] 103.7 [31.4] vs. 90.9 [27.4] beats/min, p=0.022), QTc interval (mean [SD] 470.6 [44.0] vs. 453.9 [38.7] ms, p=0.035), and LVEDD on index echocardiogram (mean [SD] 49.8 [8.0] vs. 45.6 [9.2] mm, p=0.021). Seventeen (15.2%) subjects met criteria for left ventricular (LV) dilatation, with a higher proportion of LV dilatation in *TTN*mv carriers but without a significant difference compared to non-carriers (13 [20.3%] vs. 4 [8.3%], p=0.111). Twelve subjects (9.2%) met criteria for a clinical diagnosis of nonischemic DCM based on a left ventricular ejection fraction (LVEF) <50%, presence of LV dilatation, and confirmation of non-severe coronary artery disease by coronary angiography. Of those, eight subjects carried *TTN*mvs, of which six were predicted to be deleterious (*Supplementary file 3*).

#### TTNmvs are associated with higher hospitalization risk

A total of 174 hospitalizations (64 AF-related and 110 HF-related) occurred after a median (interquartile range [IQR]) follow-up time of 4.14 (1.25–6.04) years, with 119 (68.4%) events occurring in *TTN*mv carriers and 55 (31.6%) of events occurring in non-*TTN*mv carriers. Thirty-nine (50.6%) subjects with a *TTN*mv experienced at least one hospitalization during the follow-up period compared to 22 (40.1%) of non-carriers, and total hospitalization incidence rate was 32.2 events per 100 person-years (p-y) in *TTN*mv carriers compared to 17.4 per 100 p-y in non-carriers. Mean cumulative incidence is shown in *Figure 1B*. The unadjusted hazard of hospitalization by 10 years was significantly higher in *TTN*mv carriers compared to noncarriers (hazard ratio [HR] 1.81, 95% confidence interval [CI] 1.04–3.15, p=0.036). This remained significant after partial adjustment for age and sex (HR 1.82, 95% CI 1.04–3.17, p=0.035) as well as full adjustment that additionally included ethnicity and baseline LVEF <50% (HR 1.80, 95% CI 1.03–3.15, p=0.039; *Supplementary file 4*). No other covariates were significantly associated with the outcome. To assess whether baseline LVEF influenced the relationship between *TTN*mvs and hospitalization risk, an interaction term between LVEF <50% and the presence of *TTN*mvs was tested and was not significant (interaction p=0.843). To assess whether the prediction of deleterious effect influenced the relationship between *TTN*mv presence and hospitalization risk, subgroups based on high or low REVEL score were separately examined (*Supplementary file 5*). Compared to non-*TTN*mv carriers, hospitalization risk was significantly increased in subjects carrying predicted deleterious *TTN*mvs (HR 1.92, 1.04–3.53, p=0.036). While subjects with *TTN*mvs not expected to be deleterious also had an elevated point estimate of hospitalization risk, this difference was not significant (HR 1.60, 95% CI 0.78–3.28, p=0.198). Findings remained similar with the exclusion of patients with nonischemic DCM (*Supplementary file 6*). While in a previous study, we identified

**Table 1.** Clinical characteristics of ethnic minority subjects with AF stratified by presence of rare missense TTN variants.

*Data are missing for the following variables: eGFR (1), electrocardiogram within 3 months of AF diagnosis (11), LVEDD (19), left atrial size (6), left atrial diameter (21). Left ventricular dilatation is defined as left ventricular end diastolic diameter greater than 2 standard deviations above the normal sex-specific mean value. Variants with a REVEL score ≥0.7 were defined as predicted deleterious. Continuous data are represented as mean (standard deviation) and categorical data are represented as count (%).

| | *TTN* Missense Absent (N=54) | *TTN* Missense Present (N=77) | Total (N=131) | p-value |
|---|---|---|---|---|
| Age at AF diagnosis (years) | 64.3 (15.2) | 62.9 (12.9) | 63.5 (13.8) | 0.575 |
| Male sex | 30 (55.6%) | 40 (51.9%) | 70 (53.4%) | 0.724 |
| Race/ethnicity | | | | 0.696 |
| Non-Hispanic Black | 37 (68.5%) | 56 (72.7%) | 93 (71.0%) | |
| Hispanic/Latinx | 17 (31.5%) | 21 (27.3%) | 38 (29.0%) | |
| BMI (kg/m$^2$) | 33.7 (9.0) | 34.2 (10.1) | 34.0 (9.6) | 0.765 |
| Diabetes | 18 (33.3%) | 32 (41.6%) | 50 (38.2%) | 0.366 |
| Hypertension | 45 (83.3%) | 68 (88.3%) | 113 (86.3%) | 0.448 |
| Coronary artery disease | 13 (24.1%) | 20 (26.0%) | 33 (25.2%) | 0.841 |
| History of stroke/transient ischemic attack | 8 (14.8%) | 18 (23.4%) | 26 (19.8%) | 0.270 |
| Congestive heart failure | 21 (38.9%) | 33 (42.9%) | 54 (41.2%) | 0.720 |
| Nonischemic dilated cardiomyopathy | 4 (7.7%) | 8 (11.1%) | 12 (9.7%) | 0.760 |
| Estimated glomerular filtration rate (mg/dL) | 71.8 (24.4) | 67.2 (24.7) | 69.1 (24.6) | 0.297 |
| Ventricular rate | 92.1 (29.0) | 97.5 (29.6) | 95.3 (29.4) | 0.326 |
| QRS interval (ms) | 97.6 (23.7) | 100.2 (28.1) | 99.2 (26.3) | 0.593 |
| QTc interval (ms) | 449.6 (37.7) | 466.5 (42.3) | 459.6 (41.2) | 0.027 |
| Left ventricular ejection fraction (%) | | | | 0.722 |
| Normal (≥50%) | 32 (59.3%) | 45 (58.4%) | 77 (58.8%) | |
| Mildly decreased (40–49%) | 7 (13.0%) | 7 (9.1%) | 14 (10.7%) | |
| Moderately decreased (30–39%) | 4 (7.4%) | 9 (11.7%) | 13 (9.9%) | |
| Severely decreased (20–29%) | 7 (13.0%) | 9 (11.7%) | 16 (12.2%) | |
| Very severely decreased (<20%) | 4 (7.4%) | 7 (9.1%) | 11 (8.4%) | |
| Left ventricular end diastolic diameter (mm) | 45.3 (9.2) | 47.8 (9.8) | 46.7 (9.5) | 0.180 |
| Left ventricular dilatation | 4 (8.3%) | 13 (20.3%) | 17 (15.2%) | 0.111 |
| Left atrial size | | | | 0.675 |
| Normal | 17 (32.7%) | 21 (28.8%) | 38 (30.4%) | |
| Mildly dilated | 15 (28.8%) | 23 (31.5%) | 38 (30.4%) | |
| Moderately dilated | 13 (25.0%) | 17 (23.3%) | 30 (24.0%) | |
| Severely dilated | 7 (13.5%) | 12 (16.4%) | 19 (15.2%) | |
| Left atrial diameter (mm) | 39.6 (7.4) | 41.2 (7.7) | 40.5 (7.6) | 0.286 |
| Number of TTN missense variants | | | | - |
| 0 | 54 (100.0%) | 0 (0.0%) | 54 (41.2%) | |
| 1 | 0 (0.0%) | 37 (48.1%) | 37 (28.2%) | |
| 2 | 0 (0.0%) | 26 (33.8%) | 26 (19.8%) | |
| >2 | 0 (0.0%) | 14 (18.2%) | 14 (10.7%) | |

*Table 1 continued on next page*

Table 1 continued

|  | TTN Missense Absent (N=54) | TTN Missense Present (N=77) | Total (N=131) | p-value |
|---|---|---|---|---|
| Number of predicted deleterious TTN missense variants |  |  |  | - |
| 0 | 54 (100.0%) | 34 (44.2%) | 88 (67.2%) |  |
| 1 | 0 (0.0%) | 34 (44.2%) | 34 (26.0%) |  |
| 2 | 0 (0.0%) | 9 (11.7%) | 9 (6.9%) |  |

likely pathogenic or pathogenic AF variants in 7.0% of ethnic minority probands, with most (46.7%) *TTN*tvs (*Chalazan et al., 2021*), these results suggest that *TTN*mvs are also linked with adverse clinical outcomes, and missense variants should be considered in assessing pathogenicity in clinical and genetic contexts.

### A single amino acid change in a *TTN*mv is potentially causative of AF

We identified three early-onset paroxysmal AF probands from separate families harboring the same rare *TTN*mv (Chr 2q31.2 (GRCh38): g. 178539797G>A) without additional loss-of-function *TTN* variants. The clinical characteristics of the three probands are summarized in *Table 2*. This missense variant localizes in the A-band of *TTN* and causes the substitution of a conserved residue threonine by an isoleucine (p.Thr32756Ile, NM_001267550.2 (TTN), c.98267C>T *Figure 1C–D*, *Figure 1—figure supplement 1A*). In the gnomAD reference population, the *TTN*-T32756I variant has a frequency of 0.000245 overall, 0.0046 within the subpopulation of African ancestry, and is impartially distributed across sexes and ages (*Figure 1E*, *Figure 1—figure supplement 1B–E*). The *TTN*mv has also been described in Clinvar (NM_001267550.2(TTN):c.98267C>T (p.Thr32756Ile), *Supplementary file 7Supplementary file 7*) and gnomAD v4.0.0 (SNV:2–178539798 G-A(GRCh38)) in patients with DCM and other diseases.

### *TTN*-T32756I iPSC-aCMs exhibit aberrant contractility without compromising sarcomere integrity

To functionally characterize the *TTN*-T32756I variant, we exploited wild-type (WT) human iPSC lines and introduced the point mutation c.98267C>T in the *TTN* allele (*Figure 2A*, *Figure 2—figure supplement 1A–B*). The iPSC lines expressing the pluripotent markers Sox2 and Oct4 and showing normal karyotype were first differentiated into cardiomyocytes (*Figure 2—figure supplement 1C–D*), and then a retinoic acid-based and comprehensive maturation protocol was applied to generate the iPSC-aCMs as we described previously (*Ly et al., 2021*; *Ly et al., 2022*). Given that cardiac contractility is a key component of heart function and is linked to cardiac disorders associated with *TTN* (*Schick et al., 2018*), we first assessed the contractility of the WT and mutant iPSC-aCMs and observed both reduced contraction duration and abnormal relaxation in *TTN*-T32756I-iPSC-aCMs (*Figure 2B–E*, *Figure 2—figure supplement 2A–B*). Compared to WT, the beating frequency of the *TTN*-T32756I-iPSC-aCMs was significantly increased (52±7.8 vs 98±7.5 beats per min, p=0.001; *Figure 2C*) coupled with the reduction of the contraction duration (456.5±61.45 vs 262.9±48.16 msec, p=0.032; *Figure 2D*), the peak-to-peak time (1529±195.5 vs 636.6±135.8 msec, p=0.004; *Figure 2—figure supplement 2B*), and the relaxation duration (281.5±42.95 vs 79.40±21.14 msec, p=0.003; *Figure 2—figure supplement 2A*). In contrast, the contraction amplitude of the mutant was increased (8.1±0.8 vs 10.6±7 au, p=0.040) without any significant changes in time-to-peak (*Figure 2E*, *Figure 2—figure supplement 2C*), suggesting an increased contractile force by the *TTN*-T32756I-iPSC-aCMs. As sarcomere disorganization is often observed in *TTN*-related cardiomyopathies and underlies contractile dysfunction (*Hinson et al., 2015*), we explored whether *TTN*mvs perturbed the organization of the sarcomere. Surprisingly, we did not find any disarray in the sarcomeres of either WT or *TTN*-32756I-iPSC-aCMs. Using transmission electron microscopy (TEM) and immunofluorescence (IF), we observed symmetrical arrays of recurring sarcomeres in the WT and the mutant atrial cardiomyocytes with no changes in the sarcomere length (*Figure 2F–G*, *Figure 2—figure supplement 2D–F*). Although no sarcomeric perturbation by the T32756I variant was observed, the aberrant contractility displayed by the *TTN*-T32756I-iPSC-aCMs suggests that the *TTN*mv may be pathogenic.

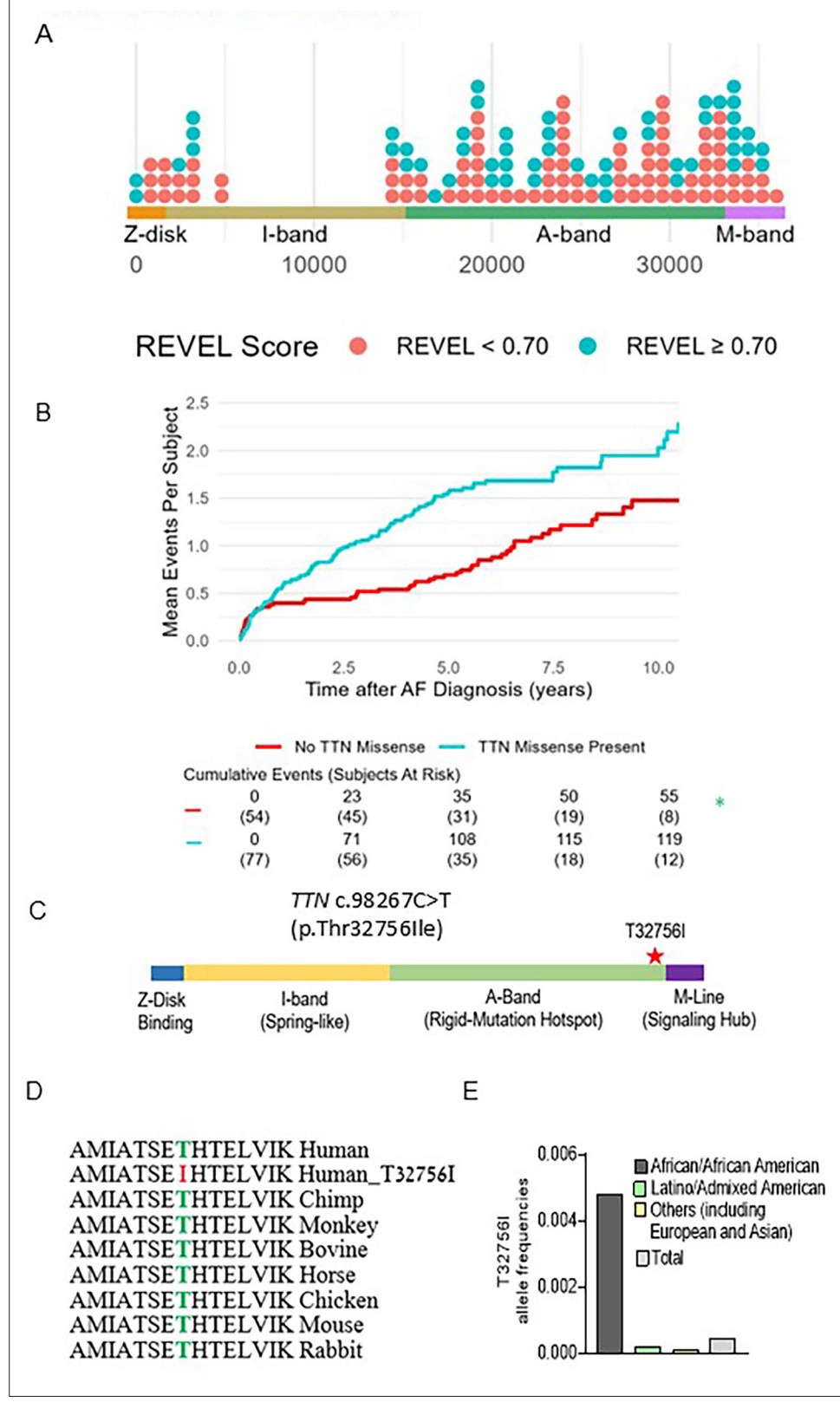

**Figure 1.** *TTN*mv prevalence and association with hospitalization in a multiethnic atrial fibrillation (AF) cohort.
(**A**) Distribution of *TTN*mvs in a multiethnic AF cohort based on amino acid position in the *TTN* gene, stratified by
REVEL in silico score for prediction of deleterious effect, defined by REVEL ≥0.70. (**B**) Mean cumulative incidence
of AF and heart failure (HF)-related hospitalizations in subjects with AF stratified by presence of *TTN*mv. Hazard

*Figure 1 continued on next page*

*Figure 1 continued*

ratio (HR), 95% confidence interval (CI), and p-value were obtained from univariable Cox proportional hazard modeling. (**C**) Diagram denoting the location of *TTN*mv-T23756I. (**D**) Sequence alignment shows that the T23756I region is highly conserved across vertebrate species. (**E**) Allele frequencies of *TTN*-T3265I in various ethnic groups (gnomAD).

The online version of this article includes the following figure supplement(s) for figure 1:

**Figure supplement 1.** *TTN*-T32756I position and distribution.

## Altered cardiac potassium current and calcium handling underlie the arrhythmia phenotype in *TTN*-T32756I-iPSC-aCMs

As studies have shown that abnormal atrial electrophysiology and calcium handling underlie atrial arrhythmogenesis and contractile dysfunction (*Nattel et al., 2020*; *McCauley et al., 2020*), we hypothesized that the *TTN*-T32756I variant modulates atrial APs, ion channels, and intracellular calcium. We first tested the spontaneous and paced AP characteristics of the isolated WT and the *TTN*-T32756I iPSC-aCMs, which revealed a significant shortening of the AP duration (APD) at 10%, 50%, and 90% repolarization; however, there was no change in the peak amplitude (*Figure 3A–C*, *Figure 3—figure supplement 1A–B*, *Figure 3—figure supplement 2A–D*). Since increased potassium current ($I_k$), especially the augmented delayed rectifier potassium current ($I_{ks}$), causes AF by reducing the APD (*Ly et al., 2022*; *Menon et al., 2019*), we assessed $I_{ks}$ by using whole-cell voltage clamping. We observed an increase in both $I_k$ and $I_{ks}$ in *TTN*-T32756I-iPSC-aCMs compared with the WT (*Figure 3D–F*, *Figure 3—figure supplement 1C–E*); thus, confirming that the APD shortening is partly due to the increased $I_k$. To test the effect of T32756I on atrial calcium handling, we measured the intracellular calcium transients at the excitation-contraction coupling moment of the iPSC-aCMs using fluorescent calcium imaging. Compared to WT, spontaneously beating *TTN*-T32756I-iPSC-aCMs exhibited increased frequency along with a significant reduction in the time to 50% and 90% decline of calcium transients (*Figure 3G–I*, *Figure 3—figure supplement 1F*). We also saw a similar decile of calcium transients in the paced iPSC-aCMs (*Figure 3—figure supplement 2E–H*). However, *TTN*-T32756I-iPSC-CMs exhibited similar calcium transient amplitudes as the WT, indicating that there was no change in the availability of the intracellular calcium for each contraction (*Figure 3—figure*

**Table 2.** Clinical characteristics of the early-onset AF patients with *TTN*-T32756I variation.

| | Case 1 | Case 2 | Case 3 |
|---|---|---|---|
| Age at diagnosis (years) | 52 | 60 | 38 |
| Sex | Male | Female | Male |
| Race/ethnicity | Hispanic | Black | Black |
| Body mass index (kg/m²) | 42.2 | 26.9 | 31.4 |
| Type | Paroxysmal | Paroxysmal | Paroxysmal |
| Comorbidities | Hypertension<br>Prostate cancer | Hypertension<br>Hyperlipidemia<br>Coronary artery disease<br>Uterine/vulva cancer<br>Severe mitral regurgitation | Hypertension<br>Asthma |
| Family history | No | No | No |
| Presenting symptoms | Asymptomatic, found during preoperative evaluation | Asymptomatic, in setting of gastrointestinal bleed | Palpitations, dyspnea |
| LA size (mm) | 51 | 33 | 35 |
| LVEF (%) | 55 | 50 | 60 |
| Antiarrhythmic | No | No | No |
| Ablation | Yes | No | No |
| Cardioversion | Yes | No | Yes |

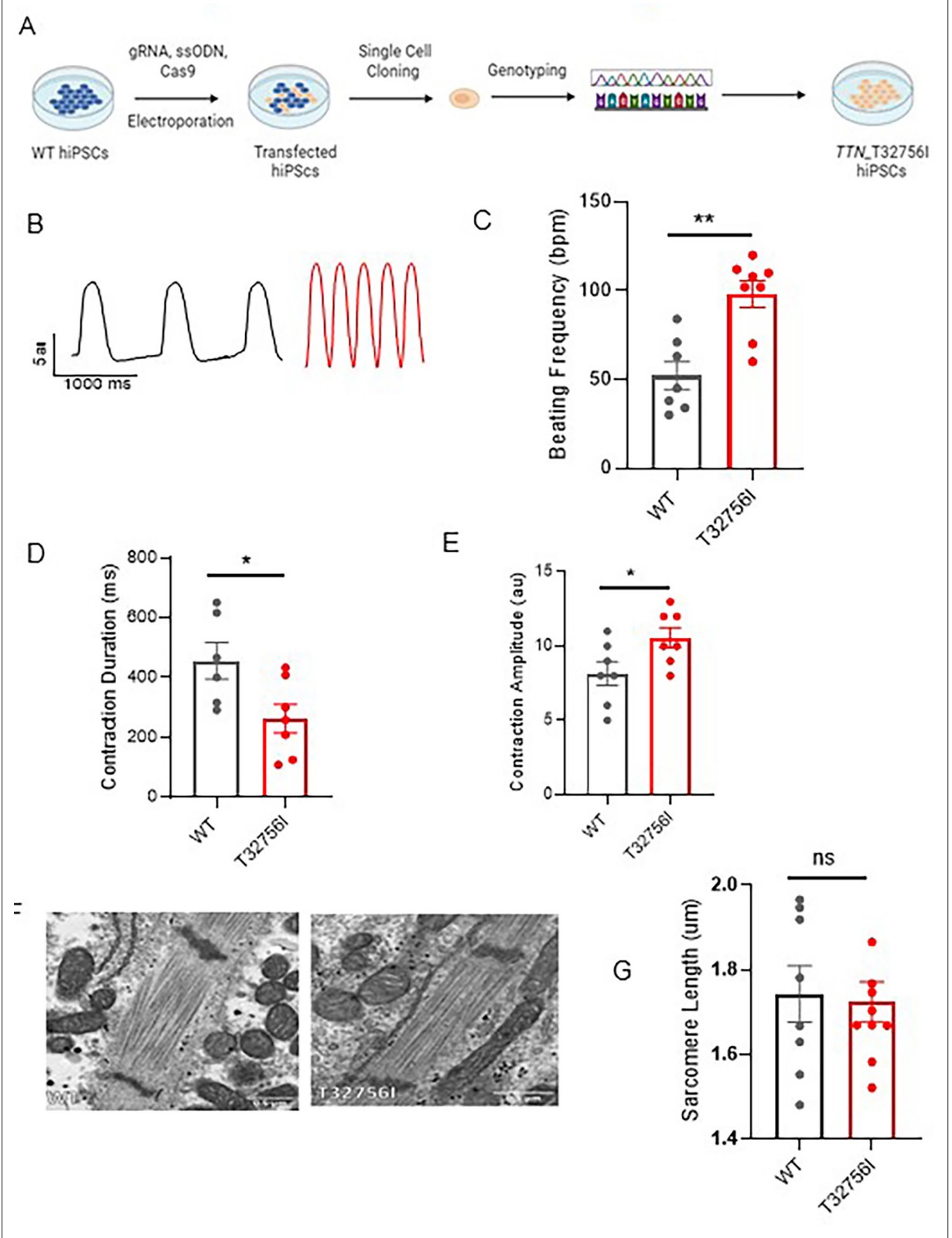

**Figure 2.** Human-induced pluripotent stem-cell-derived atrial cardiomyocytes (iPSC-aCMs) carrying TTN-T32756I variants exhibit atypical contractility but no sarcomere disorganization. (**A**) Workflow for generating the CRISPR/Cas9-mediated iPSC line carrying the *TTN*-T32756I missense variation. (**B–E**) Contraction profile of wild type (black) and *TTN*-T32756I (Red) iPSC-aCMs (**B**) showing increased beating frequency (**C**), decreased contraction

*Figure 2 continued on next page*

*Figure 2 continued*

duration (**D**), and increased contraction amplitude (**E**) in the mutant. (**F**) Representative sarcomeric organization of wild-type (WT) and *TTN*-T32756I iPSC-aCM by Transmission electron microscopy (TEM). (**G**) There is no significant change in the sarcomere length. n.s. p>0.05; *p<0.05; **p<0.01.

The online version of this article includes the following figure supplement(s) for figure 2:

**Figure supplement 1.** Generation of iPSC-aCMs with *TTN*-T32756I.

**Figure supplement 2.** Contractility and sarcomere organization of *TTN*-T32756I iPSC-aCMs.

supplement 1G, *Figure 3—figure supplement 2F*). Our findings suggest that the reduced APD resulting from increased potassium current and the altered timing of the calcium transient may create a reentrant substrate for AF.

## RNA sequencing analysis reveals alterations in cardiac signaling and disease pathways in *TTN*-T32756I-iPSC-aCMs

To elucidate the underlying molecular mechanism by which *TTN*-T32756I is associated with contractile defects and ion channel remodeling, we performed transcriptomic sequencing of mutant iPSC-aCMs and WT. Comparison of total RNA levels in *TTN*-T32756-iPSC-aCMs with WT-iPSC-aCMs and differential expression analysis showed genes related to cardiac muscle contraction and calcium handling were predominantly affected (*Figure 4A*, *Figure 4—figure supplement 1A–C*). Gene ontology (GO) pathway enrichment further showed that the top significantly downregulated cardiac-related GO Biological Processes (-BP) in *TTN*-T32756I iPSC-aCMs included key processes such as cardiac myofibril assembly, skeletal muscle myosin thick filament assembly, extracellular matrix organization, and regulation of potassium ion transmembrane transporter assembly among others (*Figure 4B*). As for GO-Molecular function pathways (GO-MF), outward potassium channel activity, voltage-gated potassium channel activity, calcium channel activity, and gap junction channel activity were enriched (*Figure 4C*). Kyoto Encyclopedia of Genes and Genomes (KEGG) pathway analysis identified downregulation in critical pathways including Adrenergic signaling in cardiomyocytes, the Phosphoinositide 3-kinase (PI3K) signaling pathway, Dilated cardiomyopathy pathway, Hypertrophic cardiomyopathy pathway, and Calcium signaling pathway (*Figure 4D*), which are crucial for normal cardiac electrophysiology and contractility. Upregulated pathways in both GO and KEGG analyses were mostly restricted to neuronal pathways, including axon and neuron development, the Hippo signaling pathway, and the Notch signaling pathway. In the transcriptomic analysis of *TTN*-T32756I-iPSC-aCMs, several transcription factors (TFs) showed significant alterations, highlighting their potential roles in the mutation's impact (*Figure 4E*, *Figure 4—figure supplement 1D*). Top-cardiac-related upregulated TFs include *MYC*, linked to cell growth and apoptosis, suggesting an increase in cellular proliferation. Stress-related TFs such as *NFE2L2* and *CEBPB* are also elevated, indicating a heightened stress response. Additionally, TFs like *ESRRA* and *FOXM1*, which regulate metabolic processes and cell cycle, respectively, and *FOXO1*, which regulates metabolism and stress resistance. Key TFs in muscle function, such as MYOD1, MYOCD, KLF5, and NKX2.5, were enhanced, alongside *SMAD2*, *CREBBP*, *PITX2*, *NFKB1*, and *GATA4*, which are vital for diverse roles ranging from signal transduction to immune regulation (*Figure 4E*). Conversely, downregulated transcription factors include *SIRT1*, which plays a role in cellular stress resistance and mitochondrial function, pointing to potential vulnerabilities in cellular defenses. Cardiovascular-related TFs such as *KLF2* and *KLF3*, along with *SMAD7*, involved in TGF-beta signaling, were reduced, possibly impacting structural and signaling integrity. Additionally, *MXD1* and *IKZF1*, crucial for cellular differentiation and immune development, were also diminished (*Figure 4E*). These transcriptional changes underscore a complex network of regulatory adjustments that could contribute to the altered EP and structural properties in *TTN*-T32756I-iPSC-aCMs.

Using the KEGG and GO pathways, we performed Ingenuity Pathway Analysis (IPA) to identify novel titin-interacting proteins that could account for the increased $I_{ks}$ activity in *TTN*-T32756I-iPSC-aCMs. The IPA suggests FHL2, which is known to transduce mechanical signaling through its interactions with titin and is enriched in the mutant iPSC-aCMs (*Figure 4F*). As FHL2 has also been shown to interact with the $I_{ks}$-binding partner mink (KCNE1) directly and increases its current density (*Kupershmidt et al., 2002*), we postulate that elevated FHL2 levels in *TTN*-T32756I-iPSC-aCMs may enhance binding to $I_{ks}$ and thus increase its activity.

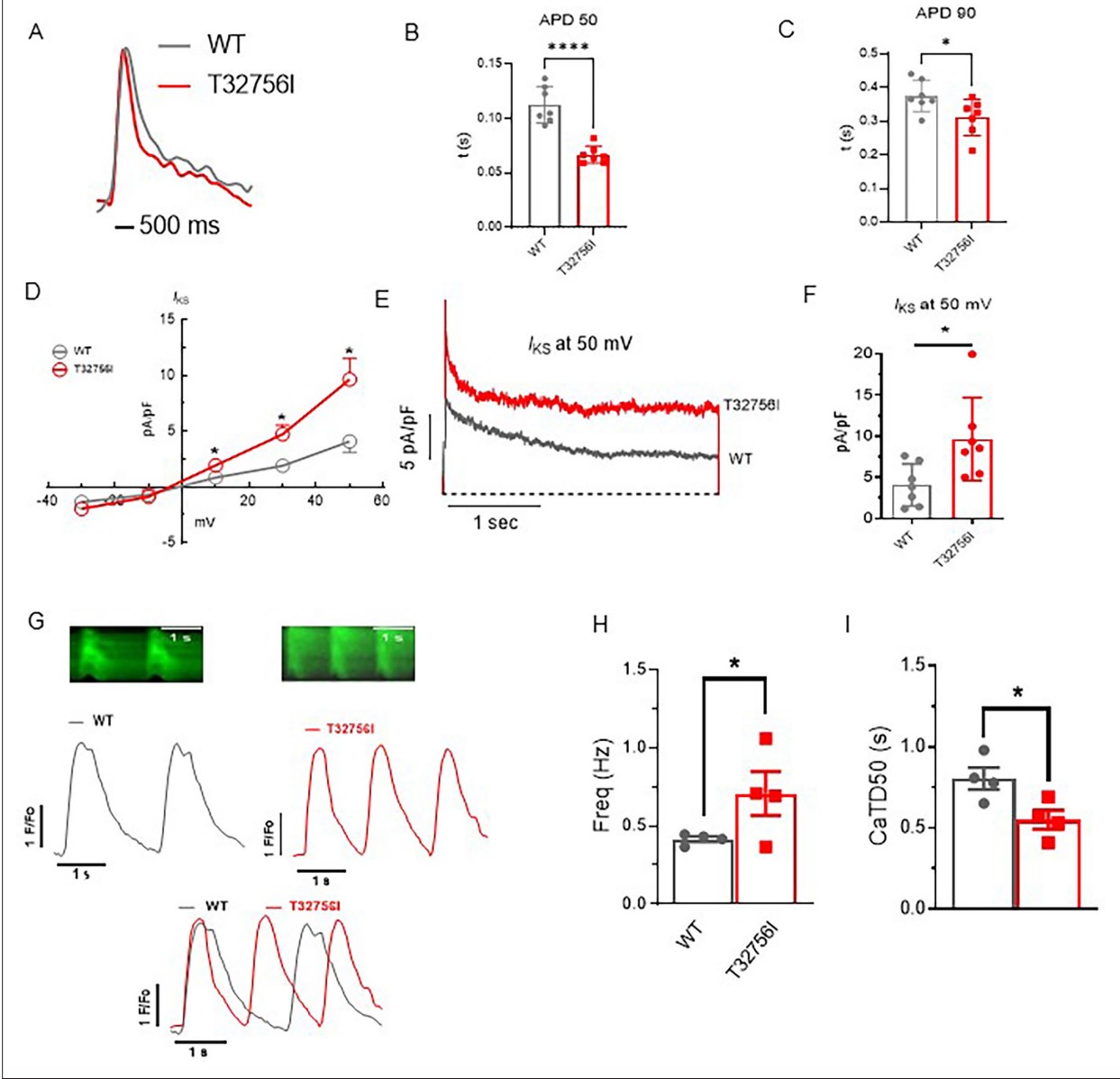

**Figure 3.** Effect of T32756I on action potential (AP) and calcium handling in iPSC-aCMs. (**A–C**) Representative optical AP recordings of spontaneously beating WT (**A**) and *TTN*-T32756I showing reduction of AP duration (APD) at the 50% (APD50) (**B**) and 90% (APD90) repolarization (**C**). (**D**) Current-voltage (*I–V*) curves of the slow delayed rectifier potassium current ($I_{ks}$) in WT and *TTN_*T32756I iPSC-aCMs (n=7). (**E–F**) Comparison of $I_{ks}$ current density at 50 mV (mean ± SEM). (**G**) Representative tracings of spontaneous calcium transients of WT and *TTN*-T32756I iPSC-aCMs. (**H–I**) Calcium kinetics show that the *TTN*-T32756I iPSC-aCMs have increased frequency (**H**) and decreased transient durations (**I**) compared with the WT iPSC-aCMs. n.s. p>0.05; *p<0.05.

The online version of this article includes the following figure supplement(s) for figure 3:

**Figure supplement 1.** *TTN*-T32756I iPSC-aCMs display anomalous action potentials, potassium currents, and calcium handling.

**Figure supplement 2.** Effect of T32756I on action potential and calcium handling in paced iPSC-aCMs.

## The FHL2 mediates the enhanced $I_{ks}$ current in *TTN*-T32756I-iPSC-aCMs

To determine if *TTN*-T32756I increases $I_{ks}$ by modulating the interaction between KCNQ1-KCNE1 and FHL2, we performed co-immunoprecipitation studies and confocal microscopy in both WT and *TTN*-T32756I-iPSC-aCMs. The co-localization between KCNE1 and FHL2 increased ~threefold in *TTN*-T32756I-iPSC-aCMs, suggesting an increased interaction between them (*Figure 5A*, *Figure 5—figure supplement 1*). Since FHL2 enhances $I_{ks}$ activity, we investigated whether inhibiting FHL2 could reverse

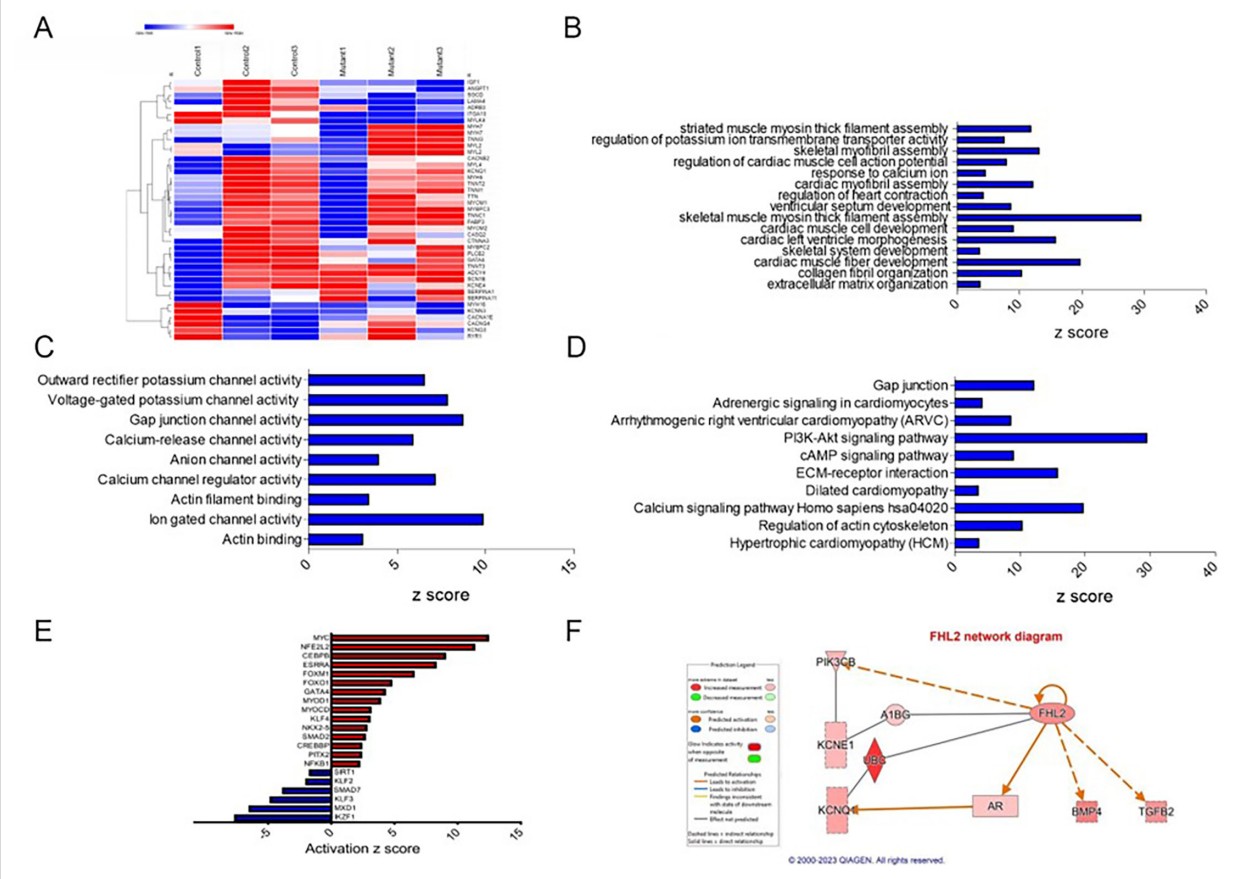

**Figure 4.** Transcriptomic profile and pathway enrichment analysis comparing *TTN*-T32756I iPSC-aCMs with the WT. (**A**) Heatmaps of cardiac-related upregulated and downregulated differentially expressed genes (DEGs). (**B**) Top significantly enriched downregulated cardiac-related Gene-Ontology Biological process (GO-BP) pathways in the *TTN*-T32756I iPSC-aCMs. (**C**) Top significantly enriched downregulated cardiac-related Gene-Ontology Molecular Function (GO-MF) pathways in the *TTN*-T32756I iPSC-aCMs. (**D**) Top significantly enriched downregulated cardiac-related Kyoto Encyclopedia of Genes and Genomes (KEGG) pathways in the *TTN*-T32756I iPSC-aCMs. (**E**) Significantly enriched upregulated and downregulated transcription factors (TFs). (**F**) Network diagram showing the upregulation of KCNQ1 by FHL2 predicted by the Ingenuity pathway enrichment analysis (IPA).

The online version of this article includes the following figure supplement(s) for figure 4:

**Figure supplement 1.** Upregulated pathways in *TTN*-T32756I iPSC-aCMs with the WT.

the increased $I_{ks}$ observed in the *TTN*-T32756I-iPSC-aCMs. To inhibit FHL2, we employed small interfering (si) RNA to suppress *FHL2* expression and measured $I_{ks}$ activity in both WT and *TTN*-T32756I-iPSC-aCMs. Both the WT and mutant iPSC-aCMs transfected with FHL2-specific siRNA showed a substantial decrease in FHL2 expression (***Figure 5B***). As shown in ***Figure 5C–D***, voltage-clamp recordings demonstrate that the $I_{ks}$ in FHL2-suppressed *TTN*-T32756I-iPSC-aCMs are significantly reduced compared to the corresponding $I_{ks}$ in *TTN*-T32756I-iPSC-aCMs but comparable to the WT (***Figure 5D***). Therefore, inhibition of FHL2 by the siRNA rescues the increased $I_{ks}$ of the *TTN*-T32756I-iPSC-aCMs, which is similar to that of WT $I_{ks}$. Overall, our data suggest that the *TTN*mv creates an EP substrate for AF by modulating $I_{Ks}$ activity, in part by increasing the interaction between the KCNQ1-KCNE1 complex and FHL2 (***Figure 5E***).

## Discussion

We identified an association between *TTN*mvs and adverse clinical outcomes in a multiethnic cohort of patients with AF and elucidated a causal mechanism by which a *TTN*mv can drive atrial remodeling and promote AF development. Although *TTN*tvs are a well-recognized cause of DCM and have been associated with early-onset AF (***Choi et al., 2018***; ***Goodyer et al., 2019***), the role of *TTN*mvs in the pathogenesis of either arrhythmia or cardiomyopathy is unknown. Here, we demonstrate that rare,

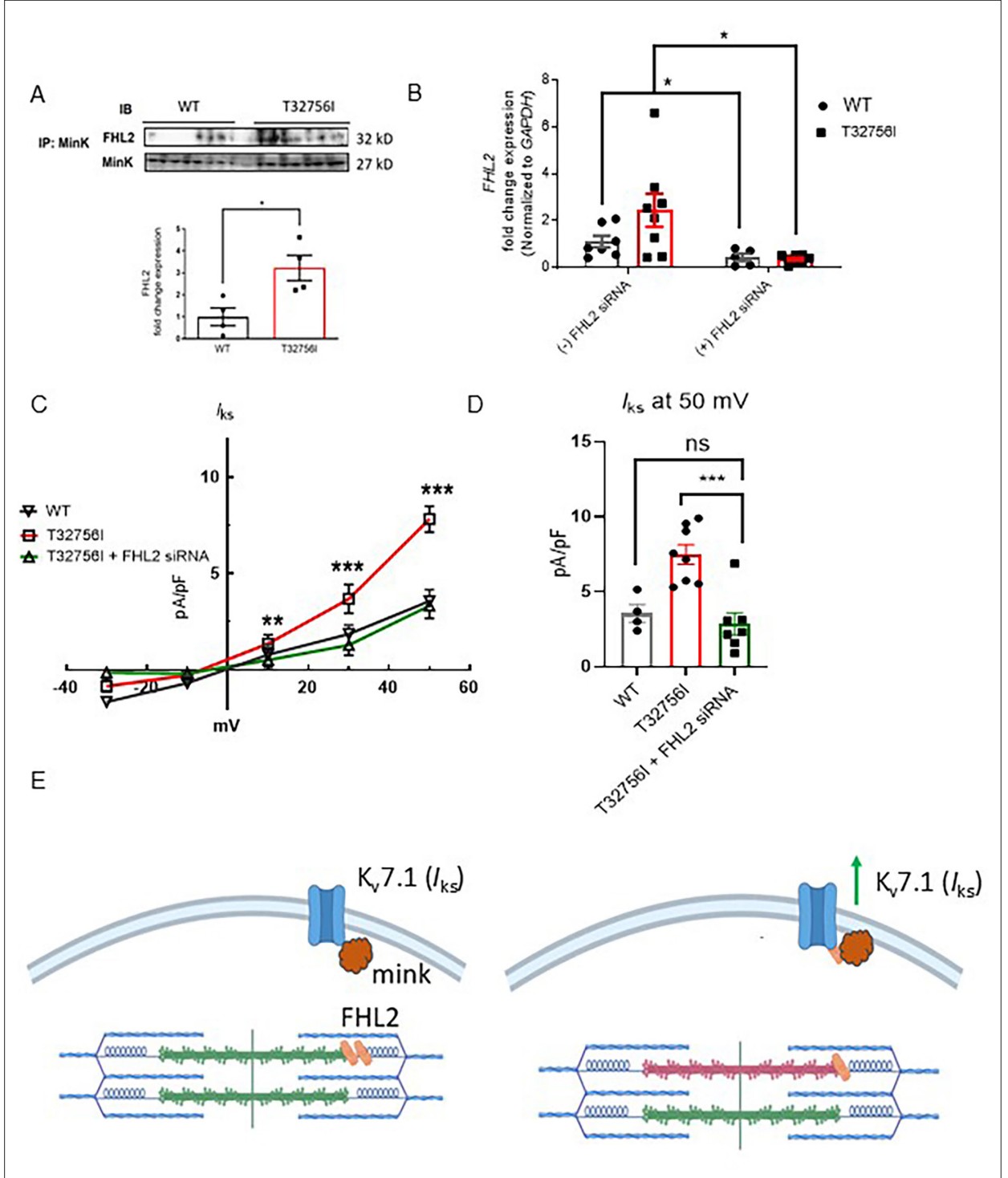

**Figure 5.** Inhibition of FHL2 rescues enhanced $I_{ks}$ in *TTN*-T32756I iPSC-aCMs. (**A**) Co-immunoprecipitation revealed increased interaction between FHL2 and KCNQ1-KCNE1 (MinK) complex. (IP: KCNE1). Immunoblotting (IB) was performed with antibodies against FHL2 (32 kDa) and MinK (32 kDa) (n=3) (**B**) qPCR data showing the inhibition of *FHL2* gene by the siRNA in the WT and *TTN*-T32756I iPSC-aCMs (n=7). (**C**) I-V curves showing the rescue of the $I_{ks}$ *TTN*-T32756I iPSC-aCMs by the suppression of FHL2 (n=4–8). (**D**) Comparison of $I_{ks}$ current density at 50 mV (mean ± SEM). (**E**) Schematic showing the *TTN*-T32756I results in increased FHL2 binding with the KNCQ1-KCNE1 complex and enhanced $I_{ks}$ activity. n.s. p>0.05; *p<0.05; ***p<0.001.

The online version of this article includes the following source data and figure supplement(s) for figure 5:

**Source data 1.** Source data contains raw images of the immunoblot presented at *Figure 5A*.

**Source data 2.** Source data contains raw images of the immunoblot presented at *Figure 5A*.

*Figure 5 continued on next page*

*Figure 5 continued*

**Figure supplement 1.** Increased FHL2-KCNE1 co-localization in TTN-T32756I iPSC-aCMs.

deleterious *TTN*mvs are associated with an elevated risk of hospitalization in a multiethnic cohort of patients with AF and show that a *TTN*mv can promote AF by impairing contractility and remodeling ion channels in atrial cardiomyocytes. Furthermore, we demonstrate that augmented $I_{Ks}$ activity mediated by FHL2 can create a substrate for AF. Together, these findings establish a mechanistic role for *TTN*mvs in AF pathogenesis and highlight $I_{Ks}$ and FHL2 as potential therapeutic targets for TTN-related arrhythmias.

The genetic architecture of AF is complex, with both monogenic and polygenic contributions that intersect with age and comorbidities to define an individual's risk. Despite AF being traditionally considered an ion channelopathy, the sarcomeric gene, *TTN,* is the most common in which rare loss-of-function variants have been associated with AF (*Choi et al., 2018*; *Yoneda et al., 2021*), and the penetrance of such variants was found to be even greater for AF than for HF in the UK Biobank (*Choi et al., 2020*). As rare *TTN* variants comprise the strongest monogenic contribution to AF risk, we sought to explore whether *TTN*mvs may similarly affect AF development or outcomes. A study of 147 probands with DCM identified 44 severe *TTN*mvs in 37 probands, which clustered in the A-band region (*Begay et al., 2015*). There were no differences in heart transplant-free survival between carriers and noncarriers. In a more recent study of 530 subjects with DCM using more stringent allele-frequency criteria, 31 predicted deleterious variants were identified, also predominantly located in the A-band (*Akinrinade et al., 2019*). However, these *TTN*mvs were found at similar frequencies to reference populations. Finally, a study of two families with DCM demonstrated segregation of *TTN*mvs affecting the same highly conserved cysteine residue in the I-band, which showed impaired contraction and folding at physiological temperatures in a homozygous iPSC model, demonstrating that a *TTN*mv can cause DCM (*Domínguez et al., 2023*).

We identified *TTN*mvs in 58% of subjects in our cohort, and 33% carried a potentially deleterious variant, as predicted by REVEL in silico. This frequency was higher than the 37/147 (25.2%) of *TTN*mv carriers in a DCM cohort (*Begay et al., 2015*), understanding that differing methods of bioinformatic filtering may limit the ability for direct comparison. Consistent with prior studies, most deleterious variants in our study were found in the A-band (*Roberts et al., 2015*; *Akinrinade et al., 2016*), where *TTN*tvs have been most strongly linked to disease. The presence of predicted deleterious *TTN*mvs was associated with a higher LV end-diastolic diameter, but without a statistically significant difference in LVEF, potentially suggesting overlap between subclinical left ventricular cardiomyopathy and AF in these subjects. Notably, rates of baseline heart failure diagnosis (encompassing both reduced and preserved ejection fraction) were similar based on *TTN*mv carrier status. We also noted higher ventricular rate and QTc interval on index ECGs in these subjects. Rare deleterious variants in *TTN* have previously been associated with changes in the QT interval (*Kapoor et al., 2016*), but the mechanism for this remains unclear. Finally, evidence on the relationship between *TTN*mvs and clinical outcomes is sparse, with only a few studies examining event-free survival in DCM patients. We observed a significant association between *TTN*mv presence and increased cumulative incidence of AF or HF-related hospitalizations, which remained unchanged in multivariable adjustment and in sensitivity analyses. This importantly suggests that *TTN*mvs may correlate with disease severity. Our cohort is derived from a single-center multi-ethnic registry of individuals with AF. It lacks a matched cohort of non-AF controls to compare the incidence of *TTN*mvs. Further study exploring these associations in multi-ethnic, larger validation cohorts that include both AF and non-AF individuals, and examining AF-specific measures such as arrhythmia burden or treatment response, will be necessary to fully understand the clinical importance of *TTN*mvs in AF.

Prediction of missense variant impact using in silico tools and algorithms can be limited by low specificity, contributing to challenges in clinical interpretation (*Richards et al., 2015*). Functional studies are often helpful in these cases to confirm or support variant pathogenicity. As existing in vitro or in vivo models do not adequately replicate the complexity of AF, human iPSC-aCMs possess the complex array of cardiac ion channels that make up the atrial AP and provide an adequate model to establish causal relationships in AF (*Ly et al., 2021*). We identified the missense variant *TTN*-T32756I in three unrelated subjects in our multiethnic AF registry. Hypertension is a common comorbidity in patients with AF and could contribute to disease progression. However, all three individuals carrying

*TTN*-T32756I exhibited early-onset AF (onset before 66 years), with one case occurring as early as 36 years. This suggests a two-hit mechanism, in which genetic predisposition and comorbidities influence disease risk. Importantly, our iPSC model isolates the genetic effects of *TTN*-T32756I from other factors, supporting a direct pathogenic role. Remarkably, a single amino acid change in the giant protein titin results in aberrant contractility and EP remodeling without affecting sarcomeric integrity. Although we did not observe any defects in the sarcomere, in a separate study, including T32756I in the same area, caused significant perturbation in sarcomere assembly (*Jiang et al., 2024*). T32756 corresponds to a conserved threonine of the Ig139 domain (UniProt# Q8WZ42, predicted Ig155 by TITINdb2), and the replacement of a conserved hydrophilic residue (Thr) in WT with a hydrophobic residue (Ile) may cause thermal instability and destabilize the domain. It is now well recognized that point mutations may drastically disrupt the Ig domains and could unfold under pathological conditions (*Li et al., 2000*). Such mutations have been proposed to modify titin-based passive stiffness and may lead to cardiomyopathy. Missense mutations such as T2850I, R57C, and S22P in the I-band titin have been reported to significantly destabilize Ig domains and display a higher tendency to unfold (*Zuo et al., 2021*; *Anderson et al., 2013*). Likewise, T32756I may unfold under the physiological settings and increase susceptibility to protein loss (haploinsufficiency) or degradation (poison peptide effect), leading to impaired sarcomere function. Furthermore, as titin acts as a crucial signaling hub that transduces mechanical forces to downstream signaling pathways within cardiomyocytes and interacts with various sarcomeric proteins and signaling molecules (*Granzier and Labeit, 2004*; *Krüger and Linke, 2011*), subtle destabilization of the Ig139 domain by the T32756I could alter the biochemical signals that regulate various downstream signaling pathways involved in cardiac function and adaptation. Hence, the contractile and ion channel dysfunction caused solely by homozygosity for T32756I in iPSC-aCMs suggests that the *TTN*-T32756I missense variant can drive atrial remodeling. While our CRISPR-edited isogenic *TTN*-T32756I iPSC-aCMs provide genetically controlled for minimizing the genetic background variability and dissecting variant-specific effects, future studies using multiple iPSC backgrounds and patient-derived lines will be critical for confirming the robustness and generalizability of the observed phenotypes.

Although several studies have identified possible pathways associated with *TTN*tvs and AF, the pathogenic mechanisms of *TTN*mv-associated AF remain unclear. We show for the first time that the T32756I creates an EP substrate for AF by ion channel remodeling with an enhanced $I_{Ks}$, which is mediated by a titin-interacting protein FHL2. FHL2 is enriched in cardiac muscle and is a multifunctional protein that regulates cardiac myocyte signaling and function (*Tran et al., 2016*). FHL2 interacts with KCNE1, which binds to the outer face of the KCNQ1 channel pore domain and may modify the interactions between the voltage sensor, S4-S5 linker, and the pore domain to augment the channel current (*Kupershmidt et al., 2002*). It has been suggested that FHL2 interacts with the titin N2B segment, titin kinase domain, and forms a complex with other proteins, such as MURF1 and MURF2, calmodulin, and Nbr1, playing a role in hypertrophy signaling and the atrophy response (*Sun et al., 2024*; *Liang et al., 2018*). It also binds to the edge of titin (M-line), which is in close proximity to T32756I and interacts with myospryn and obscurin (*Henderson et al., 2017*). It is tempting to postulate that T32756I may interfere with FHL2's interaction with titin, thereby disrupting subsequent mechano-biochemical signaling and increasing FHL2's availability for binding to the KCNQ1-KCNE1 complex. While our study provides mechanistic insights into the role of the *TTN*-32756I missense variant in AF, *TTN*mvs may modulate ion channels and regulate atrial rhythmicity through multiple interrelated signaling pathways. Further studies are needed to fully uncover the molecular basis of this regulation and its implications for AF pathophysiology and potential therapeutic interventions.

In summary, this study demonstrates an association between *TTN*mvs and AF, suggesting that *TTN*mvs may contribute to adverse clinical outcomes in patients with AF and establishing a mechanism by which a particular *TTN*mv may influence AF development. We identify a critical role for FHL2 in modulating the atrial action potential by interacting with the Kv7.1–KCNE1 complex, thereby enhancing $I_{Ks}$ in *TTN*-T32756I iPSC-aCMs. Importantly, suppression of FHL2 rescued the augmented $I_{Ks}$, indicating that targeting FHL2 or $I_{Ks}$ may help restore sinus rhythm and improve atrial contractility. These findings provide mechanistic insight into how *TTN*mvs can promote arrhythmogenesis and highlight KCNQ1 and FHL2 as potential therapeutic targets for TTN-related AF. Further studies are warranted to validate this mechanism and to determine whether *TTN*mvs in other titin domains induce AF through similar pathways. Although the effects of *TTN*-T32756I in iPSC-derived ventricular

cardiomyocytes remain to be explored, future work may clarify whether this variant produces comparable or distinct EP alterations. Collectively, our findings broaden the understanding of *TTN*-related atrial pathophysiology, underscore the importance of incorporating *TTN*mvs into genomic analyses of AF, and emphasize the value of functional validation in advancing precision medicine for patients carrying rare sarcomeric variants.

## Methods

### Study population

The UIC Multi-Ethnic AF Biorepository was established in 2015 to explore the genetic basis of AF across racial and ethnic groups. This study was approved by the UIC Institutional Review Board (#2015–0681). Subjects are prospectively enrolled from outpatient and inpatient sites within the University of Illinois Health (UIH) healthcare system. Subjects must have a documented history of AF by electrocardiogram, Holter/event monitor, or implantable cardiac device, with confirmation by an attending cardiologist. At the time of enrollment, blood is drawn for DNA extraction and genetic analysis. Baseline demographics are obtained from provider notes in the electronic health record (EHR) and, as necessary, confirmed with the patient at the time of enrollment. Longitudinal clinical outcomes are followed through serial review of the EHR. Written informed consent was obtained from all participants. Adults 18 years of age or older at the time of AF diagnosis were prospectively enrolled between August 25, 2015, and May 19, 2019. Samples from 161 subjects who identified as non-Hispanic Black (NHB) or Hispanic/Latinx (HL) and had an echocardiogram performed within 3 months of enrollment date underwent whole exome sequencing. A total of 27 subjects with congenital or rheumatic disease, severe mitral stenosis, and end-stage renal disease on dialysis were excluded. Three subjects who carried a predicted loss-of-function frameshift or stop-gain *TTN* variant were additionally excluded. The primary exposure was the presence of one or more missense *TTN* variants. The primary outcome was time from initial AF diagnosis to hospitalizations with a primary diagnosis of either AF or acute decompensated HF recorded in the clinical discharge summary, captured as repeating events. Clinical data was obtained through manual review of EHR and comorbidities were defined if present in clinical notes on or prior to AF diagnosis date. Subjects were censored on the date of death, last known clinical encounter (up to August 26, 2023), or at 10 years after AF diagnosis. Echocardiographic criteria for left ventricular dilatation were defined as left ventricular end diastolic diameter (LVEDD) >2 standard deviations (SD) above sex-specific mean according to standard guidelines (*Lang et al., 2015*). Subjects meeting echocardiographic criteria for left ventricular dilatation fulfilled clinical criteria for nonischemic dilated cardiomyopathy if left ventricular ejection fraction (LVEF) <50% and ischemic cardiomyopathy was ruled out by coronary angiogram.

### Whole exome sequencing

Samples were sequenced with support of the National Human Genome Research Institute (NHGRI) Centers for Common Disease Genetics (CCDG) program. Sequencing was performed at the Broad Institute following methods established by the National Heart, Lung, and Blood Institute (NHLBI) Trans-Omics for Precision Medicine (TOPMed) Atrial Fibrillation Study (available under TOPMed Whole Genome Sequencing Methods: Freeze 9 here). Samples underwent WES using an Illumina HiSeq X system with 150 bp paired-end read length, at 20 x depth for at least 85% of targets. Alignment to human genome reference GRCh38/hg38 was performed using BWA-MEM (Burrows-Wheeler Aligner, v0.7.15.r1140). *TTN* variant calls were filtered to select those with read depth of ≥20 X, genotype quality ≥20, minor allele frequency (MAF) ≤1% across gnomAD subpopulations, and excluding intronic, regulatory, or UTR 3'/5' variants. Synonymous variants, homozygous reference calls, noncanonical variants, and any variants with percent spliced in (PSI) index <90 were excluded. Missense variants predicted to be deleterious were identified using REVEL score of ≥0.7, a previously proposed cutoff to predict deleterious effect in dilated cardiomyopathy (*Ioannidis et al., 2016*; *Morales et al., 2020*). Because of the uncertain role of missense *TTN* variants across AF and other cardiac diseases, we examined all *TTN* missense variants regardless of REVEL score. Variant annotation data were obtained from Ensembl Variant Effect Predictor (*McLaren et al., 2016*), downloaded January 11, 2024, and CardioDB (https://cardiodb.org/titin) for *TTN* region and percent spliced in (PSI) (*Roberts et al., 2015*), downloaded November 16, 2023. The TTN-T32756I variant (REVEL Score: 0.58758;

*Supplementary file 1*) was prioritized because it occurred in multiple unrelated individuals within our clinical AF cohort, despite no reported family history of AF in affected individuals. While no parental inheritance was observed, the possibility of de novo origin cannot be excluded. Furthermore, this variant is located within a region overlapping a deletion mutation recently shown to cause AF in a zebrafish model, supporting its potential pathogenicity (*Jiang et al., 2024*). Notably, the affected individuals did not carry additional loss-of-function TTN variants.

## Human iPSC culture and human iPSC-aCM differentiation

Human iPSC-CMs were derived from reprogrammed peripheral blood mononuclear cells (PBMCs) as previously described (*Hong et al., 2021*). Mycoplasma testing revealed no contamination. 80–90% confluent iPSC-CMs were differentiated using the Cardiomyocyte Differentiation Kit (Gibco) and guided toward the atrial subtype using all-trans RA (*Argenziano et al., 2018*). The cellular population was purified through glucose starvation and lactate replacement, resulting in contracting monolayers of iPSC-aCMs. Our protocol typically yields ~80 to 90% pure iPSC-aCMs and <6% fibroblasts based on immunostaining analysis as we have previously described (*Hong et al., 2021*; *Argenziano et al., 2018*). Human iPSC-aCMs were then matured following dissociation and replating on fibronectin-coated plates and maintained in Cardiomyocyte Maintenance Media supplemented with T3, insulin-like growth factor-1, and dexamethasone as previously described (*Ly et al., 2022*).

## *Generation of TTN-T32756I hiPSCs using CRISPR-Cas9*

Due to the patients' unavailability of peripheral blood mononuclear cells (PBMCs), we utilized a healthy iPSC line and introduced the TTN-T32756I variant using CRISPR/Cas9 genome editing. This approach ensures an isogenic background, thereby minimizing genetic variability and providing a controlled system to study the direct effects of the mutation. We edited the genome of WT iPSCs to introduce the T32756I using the CRISPR-Cas9 technique (*Hong et al., 2021*). The *TTN*-T32756I CRISPR-Cas9 was designed from WT allele sgRNA and two ssODN that target the *TTN* genomic locus: 31111–31119 and genomic DNA location: chr2:178539785–178539811 (GRCh38.p13). An amplification of the exon 352 cDNA donor RNP was constructed and electroporated into *TTN* gene WT hiPSCs. Gene-editing efficiency was confirmed with next-generation sequencing (NGS). Karyotype analysis was carried out by the Cytogenetics laboratory at WiCell Research Institute Inc Cells were collected and chromosomes were evaluated using the Giemsa trypsin Wright (GTW) banding method. Metaphase cells were analyzed, all of which were concluded to have a normal karyotype (46, XY).

## Contraction analysis and optical voltage mapping

Contractility of hiPSC-aCMs was performed with a versatile open-source software, MUSCLEMOTION (*Sala et al., 2018*). MUSCLEMOTION is an ImageJ plugin that utilizes a video-based system to assess contractile functions. Optical voltage mapping recordings were performed on the IonOptix system myopacer system using the fluovolt membrane potential kit (Thermo Fisher). HiPSC-aCMs were cultured in confocal dishes and were incubated with Tyrode's solution (140 mM NaCl, 4.56 mM KCl, 0.73 mM $MgCl_2$, 10 mM HEPES, 5.0 mM dextrose, 1.25 mM $CaCl_2$) plus 1 x Fluovolt (Sigma/Aldrich) for 15–20 min. Cells were then rinsed with normal Tyrode's solution and recorded APDs on the IonOptix system. For the pacing experiment, iPSC-aCMs were electrically stimulated at a frequency of 0.5 Hz with platinum wires separated 2–4 mm, and stimulation voltage was 25% above threshold (10 V, 4 ms). These experiments were performed at room temperature (25 °C).

## Electrophysiology

Whole-cell patch clamping on hiPSC-aCMs $I_K$ recordings was performed according to previously published protocols (*Ly et al., 2022*; *Hong et al., 2021*). Briefly, voltage clamps were achieved by using an Axopatch 200B amplifier controlled by pClamp10 software through an Axon Digidata 1440 A. External solution for $I_K$ recordings contained: 140 mM NaCl, 4 mM KCl, 1.8 mM CaCl2, 1.2 mM MgCl2, 10 mM glucose, 10 mM HEPES, and 0.01 mM nifedipine, adjusted to pH 7.4 with NaOH. $I_{Ks}$ recordings were isolated as 1 μM HMR-1556–sensitive current. The intracellular solution contained 100 mM potassium aspartate, 2 mM MgCl2, 20 mM KCl, 5 mM Mg-ATP, 5 mM EGTA, and 10 mM HEPES adjusted to 7.2 with KOH. $I_{Ks}$ currents were elicited by using 3 s voltage-clamp steps to test potentials of −60 to +60 mV from holding potential of −40 mV and with 20 mV increments.

## Calcium handling

Calcium transients were measured using Fluo-4-AM (Invitrogen) dye dissolved in 2.5% Pluronic F-127 (MilliporeSigma). To get a working concentration of 5 µM, the dye solution was added to Tyrode's solution (1 mM $Ca^{2+}$). The cells were treated with Fluo-4-AM in 1 mM $Ca^{2+}$ Tyrode's solution, and then allowed to sit at room temperature for 20 min in the dark. Tyrode's solution without indicators and containing 2 mM calcium was used to wash the cells. Using a 40× objective and the Zeiss LSM 710 confocal equipped with a BiG module, line scans were acquired and examined using ImageJ. The corrected minimum and maximum fluorescence values were determined by normalizing the fluorescence using a baseline background region that was unique to each cell. Fluorescence signals were normalized to basal cell fluorescence after fluo-4 loading (F0). For the pacing experiment, iPSC-aCMs were incubated at room temperature with 1 µM Fura 2-AM for 10 min and then washed out with normal tyrode 1.8 mM Ca for 20 min. The cells were then electrically stimulated at a frequency of 0.5 Hz with platinum wires separated 2–4 mm; stimulation voltage was 25% above threshold (10 V, 4 ms). Estimations of intracellular $Ca^{2+}$ are reported as changes in ΔF/F0, where $\Delta F$=F-F0. Calcium transients' amplitude 340/380 nm ratio, and kinetics were analyzed using IonOptix software.

## Transmission electron microscopy (TEM)

iPSC-aCMs were fixed with 2.5% glutaraldehyde in 0.1 M Sorenson's Buffer for 60 min at room temperature. The cells are then carefully collected into a microcentrifuge tube pre-filled with the same fixation buffer. The samples were then centrifuged at 2500 x $g$ for 10 min at room temperature. The pellet was then removed and inverted with a hypodermic needle to ensure fixative solution thoroughly permeates the sample. The pellet is then allowed to incubate at room temperature to fix for an additional 60 min at room temperature. The fixation solution was then removed and substituted with 1% glutaraldehyde +4% paraformaldehyde in 0.1 M Sorenson's buffer and stored in 4 °C. Fixed samples were embedded in resin and microtome sections were imaged on a JEOL JEM-1400 Flash TEM.

## Immunofluorescence

Cells were cultured on Matek(R) glass bottom dishes that were coated with vitronectin. The cells were then allowed to grow for a period of 2 days. The cells were then fixed with 4% paraformaldehyde (PFA) for 10 min and permeabilized with phosphate-buffered saline (PBS) containing 0.1% Triton X-100 for 10 min. After that, the cells were blocked with 3% bovine serum albumin (BSA) in PBS for 1 hr at room temperature. The staining procedure was carried out overnight at 4 °C using primary antibodies that were diluted in 3% BSA in PBS. The cells were rinsed three times with PBS and then exposed to secondary antibodies (anti-mouse-FITC or anti-rabbit-PE, Santacruz, 1:1000) for 1 hr at room temperature. Subsequently, the cells were rinsed 3 x with PBS and co-stained with 1 µg/ml of DAPI. The antibodies Anti-SOX2 (Abcam, 1:200) and anti-OCT4 (Santacruz, 1:200) were used to assess the generation of iPSCs, while anti-cTnT (Invitrogen, 1:300) and anti-Kv1.5 (Alomone labs, 1:200) were utilized to confirm the differentiation of atrial cardiomyocytes. The samples were visualized using an LSM710 Meta Confocal Microscope manufactured by Zeiss. For the co-localization experiment, anti-FHL2 antibody (Abcam, #ab202584) and anti-KCNE1 antibody (Alomone Labs, #APC-163) were used and the subsequent captured images were analyzed using ImageJ software with the Coloc2 plugin. Pearson's correlation coefficient was calculated to quantify the degree of co-localization between FHL2 and KCNE1 signals.

## Protein isolation, western blotting, and co-immunoprecipitation

We performed Western blots as previously described (*Ly et al., 2022*; *Hong et al., 2021*). For western blots, cells on six-well plates were washed with ice-cold DPBS without $Ca^{2+}$ and $Mg^{2+}$, after which 250 µL of 1 X RIPA with protease and phosphatase inhibitors was added per well. Lysate concentrations were measured using BCA Assay and diluted with 4 X Laemmli buffer with 10% 2-Mercaptoethanol. Per sample, 25 µg of protein was then run on an SDS-polyacrylamide gel to separate, and resolved gels were electro-transferred on 0.2 µm PVDF membranes. Membranes were blocked with 5% BSA for 1 hr and then probed with corresponding antibodies of target proteins (anti-FHL2 antibody, Abcam#ab202584). The blots were developed using either anti-rabbit HRP or anti-mouse HRP and scanned on C280 imaging systems (Azure Biosystems). Protein signal densities were determined using ImageJ and normalized to corresponding β-actin signal densities.

For co-immunoprecipitation experiments, cells are rinsed with ice-cold DPBS without $Ca^{2+}$ and $Mg^{2+}$, after which 500 µL of 1 X RIPA with protease and phosphatase inhibitors was added to each well of a six-well plate. Lysates are sonicated on ice 3x5 s, then centrifuged for 10 min at 14,000 x $g$ at 4 °C. 100 µL of Protein A/G agarose bead slurry was added to the lysate and incubated at 4 °C on a rotator for 60 min. The cell lysate with bead slurry was centrifuged for 10 min at 1000 x $g$ at 4 °C and the supernatant was transferred to a fresh Eppendorf tube. 2 µg of primary antibody (Anti-KCNE1 [Alomone# APC-163]: 2.5 µL) was added to 500 µg of precleared cell lysate, and the mixture was incubated with gentle rotation overnight at 4 °C for a day. 40 µL of Protein A/G bead slurry was washed 3 x with RIPA, and the precleared cell lysate with the primary antibody was added to the bead slurry, and then followed by incubation for 3 hr at 4 °C. The mixture is centrifuged for 10 min at 1000 x $g$ at 4 °C, and the flow through was saved to verify the immunoprecipitation. The bead pellet was rinsed with 5x500 µL RIPA and resuspended in 30 µL 4 X Laemmli sample buffer, vortexed, then centrifuged for 30 s. The sample was then boiled for 5 min, then centrifuged at 14,000 x $g$ for 5 min, representing the IP sample. 15 µL of supernatant and immunoprecipitate samples were loaded onto a 10% SDS-PAGE gel and analyzed by western blotting.

## RNA sequencing

The RNA was isolated using the Maxwell RSC simplyRNA Cells Kit (Promega AS1390) according to the instructions provided by the manufacturer. Library preparation was carried out using the Universal Plus mRNA-Seq kit (NuGen 0520-A01). Briefly, RNA underwent poly-A selection, enzymatic fragmentation, and generation of double-stranded cDNA using a mixture of Oligo (dT) and random priming. The cDNA was subjected to end repair, ligation of dual-index adaptors, strand selection, and 15 cycles of PCR amplification. The concentrations of the purified library were determined using the Qubit 1 X dsDNA HS Assay Kit (Invitrogen Q33231). The libraries were then combined in equal amounts, considering the Qubit concentration and the average size determined by TapeStation. The pooled libraries were subsequently run on the MiniSeq instrument for index balancing. The ultimate, refined pool was measured using quantitative polymerase chain reaction (qPCR). The raw reads were aligned to the reference genome hg38 using the STAR alignment tool. The quantification of ENSEMBL gene expression was performed using FeatureCounts. The edgeR package was used to calculate normalized and differential expression statistics. To account for multiple testing, p-values were adjusted using the false discovery rate (FDR) correction method. We detected differentially expressed genes (DEGs) with a q value of less than 0.05 and a log 2 fold change of at least 2 between the Control and AF iPSC-aCMs. We conducted unsupervised hierarchical clustering of all genes that showed differential expression using the Euclidean distance and complete linkage method. Additionally, we created volcano plots using the R programming language. The up- and down-regulated genes were analyzed individually using the DAVID functional annotation tool against the Gene Ontology Biological Process (GO BP) database.

## qPCR

Total RNA was isolated from hiPSC-aCMs using TRIzol reagent (Invitrogen), following the manufacturer's instructions to ensure the extraction of high-quality RNA. The concentration and purity of the isolated RNA were carefully assessed using a NanoDrop 2000 spectrophotometer (Thermo Fisher Scientific), with 1 µg of total RNA utilized for each reverse transcription reaction. Reverse transcription to synthesize cDNA was conducted using SuperScript III Reverse Transcriptase (Thermo Fisher Scientific). For the analysis, specific assays and primers were selected for target gene FHL2 (Forward sequence: GTGGTGTGCTTTGAGACCCTGT, Reverse sequence: GAGCAGTGGAAACAGGCTTC ATG) with glyceraldehyde 3-phosphate dehydrogenase (GAPDH) serving as the normalization reference gene. qPCR reactions were performed on an ABI QuantStudio 5 system (Applied Biosystems), using SYBR Green PCR Master Mix to accurately detect and quantify PCR amplification products. Relative expression levels of the target genes were calculated employing the ΔΔCt method, by the quantification of gene expression changes in the experimental samples relative to control. For the $\Delta CT$ Calculation, the cycling time (CT) value of the target gene was subtracted from the CT value of GAPDH in the same sample using $\Delta CT = CT_{target\ gene} - CT_{reference\ gene}$. The ΔΔCT value was then calculated using $\Delta\Delta CT = \Delta CT_{Experimental} - \Delta CT_{Control}$. The relative expression for the gene was in turn calculated using Relative gene expression = $2^{-\Delta\Delta CT}$.

### *siRNA* experiments

FHL2-specific siRNAs (#SR301594) or scrambled siRNAs were applied to mature hiPSC-aCMs using Lipofectamine RNAiMAX (Invitrogen, Carlsbad, CA). Opti-MEM medium was used to dilute stock solutions of lipofectamine and siRNA, each prepared at a concentration of 10 µM. The siRNA-lipid complex was then made by mixing these solutions in a 1:1 ratio and incubated for five minutes. After adding this complex to the cells dropwise, the media was changed, and the cells were allowed to incubate for a further two days.

### Data analysis and statistics

For clinical data, categorical variables are represented as count and percentage (%) and tested by Fisher's exact test. Continuous variables are reported as mean (standard deviation [SD]) or median and interquartile range (IQR) where specified and tested with Mann-Whitney U and Kruskal-Wallis tests. Ordinal variables were tested with the Kruskal-Wallis test. Univariable and multivariable Cox proportional hazards models were used to evaluate risk of the primary outcome as recurring events, related to presence of *TTN* missense variants. A partially adjusted model was first tested using covariates of age and sex, followed by a fully adjusted model which additionally incorporated covariates of ethnicity and baseline LVEF <50%. Interaction between *TTN* missense variants and LVEF was additionally evaluated. To identify whether associations were limited to variants with predicted deleterious effect, analysis was repeated stratifying subjects according to presence and predicted impact of *TTN*mv: presence of one or more predicted deleterious (REVEL ≥0.7) variants, presence of predicted benign (REVEL <0.7) variants only, or no *TTN*mv present. Additional sensitivity analyses were performed excluding subjects with multiple *TTN*mvs and with nonischemic dilated cardiomyopathy. Analysis was performed in R (version 4.2.1, packages: *arsenal, ggplot2, ggsurvfit, gtsummary, reda, survival*). Experiments were performed at least two times (biological replicates) to ensure reproducibility. Unless otherwise noted, experimental data on hiPSCs are shown as mean ± SD. *p<0.05, **p<0.01, ***p<0.001, ****p<0.0001 indicate significance, while p>0.05 is regarded as non-significant. Statistical analyses include unpaired nonparametric and two-tailed Mann-Whitney U test for data with normal distribution, and either one-way or two-way ANOVA with post hoc Bonferroni's corrections for several groups. The first and third quartiles of a median are used to express skewed data. Fisher's Exact test is used to compare categorical data, and unpaired Student's t-test or ANOVA is applied to assess continuous variables.

## Acknowledgements

This work was in part supported by NIH grants R01 HL150586 (DD), R01 HL148444 (DD), NIH T32 HL139439 and AHA 23POST1019044 (MH), UL1 TR002003 (The University of Illinois Chicago Center for Clinical and Translational Science (CCTS)), AHA CDA (MAP), and UIC seed funding (MAP).

## Additional information

### Competing interests

Jalees Rehman: Reviewing editor, *eLife*. The other authors declare that no competing interests exist.

### Funding

| Funder | Grant reference number | Author |
| --- | --- | --- |
| National Heart Lung and Blood Institute | | Dawood Darbar |
| American Heart Association | | Mahmud Arif Pavel |
| NIH | R01 HL150586 | Dawood Darbar |
| NIH | R01 HL148444 | Dawood Darbar |
| NIH | T32 HL139439 | Dawood Darbar |

| Funder | Grant reference number | Author |
| --- | --- | --- |
| American Heart Association | 23POST1019044 | Michael Hill |
| AHA CDA | | Mahmud Arif Pavel |
| UIC seed funding | | Mahmud Arif Pavel |

The funders had no role in study design, data collection and interpretation, or the decision to submit the work for publication.

## Author contributions
Mahmud Arif Pavel, Dawood Darbar, Conceptualization, Resources, Data curation, Software, Formal analysis, Supervision, Funding acquisition, Validation, Investigation, Visualization, Methodology, Writing – original draft, Project administration, Writing – review and editing; Hanna Chen, Abhinaya Baskaran, Aylin Ornelas Loredo, Data curation, Formal analysis, Investigation; Michael Hill, Data curation, Software, Formal analysis, Validation, Visualization, Writing – original draft, Writing – review and editing; Arvind Sridhar, Investigation, Methodology; Miles Barney, Data curation, Formal analysis, Investigation, Visualization; Jaime DeSantiago, Data curation, Formal analysis, Investigation, Visualization, Methodology; Asia Owais, Investigation; Shashank Sandu, Faisal A Darbar, Data curation, Formal analysis; Bahaa Al-Azzam, Formal analysis, Investigation; Brandon Chalazan, Resources, Data curation, Formal analysis; Jalees Rehman, Conceptualization, Resources, Supervision, Project administration

## Author ORCIDs
Mahmud Arif Pavel (iD) https://orcid.org/0009-0005-2813-315X
Michael Hill (iD) https://orcid.org/0000-0002-5973-0804
Jalees Rehman (iD) https://orcid.org/0000-0002-2787-9292
Dawood Darbar (iD) https://orcid.org/0000-0002-4103-5977

## Ethics
This study was approved by the UIC Institutional Review Board (#2015-0681). Subjects are prospectively enrolled from outpatient and inpatient sites within the University of Illinois Health (UIH) healthcare system. Written informed consent was obtained from all participants.

Reviewer #1 (Public review): https://doi.org/10.7554/eLife.104719.3.sa1
Reviewer #2 (Public review): https://doi.org/10.7554/eLife.104719.3.sa2
Author response https://doi.org/10.7554/eLife.104719.3.sa3

---

# Additional files

## Supplementary files
Supplementary file 1. List of *TTN* missense variants. Age represents patient's age at AF diagnosis in years. M=male, F=female, HL = Hispanic/Latinx, NHB = non-Hispanic Black. Variants with a blank value in the dbSNP or gnomAD columns represent variants not present in those respective databases.

Supplementary file 2. Clinical characteristics of ethnic minority subjects with AF stratified by presence of predicted deleterious rare missense *TTN* variants. *Data are missing for the following variables: eGFR (1), electrocardiogram within 3 months of AF diagnosis (11), LVEDD (19), left atrial size (6), left atrial diameter (21). Left ventricular dilatation is defined as left ventricular end diastolic diameter greater than 2 standard deviations above the normal sex-specific mean value. Variants with a REVEL score ≥0.7 were defined as predicted deleterious. Continuous data are represented as mean (standard deviation) and categorical data are represented as count (%)

Supplementary file 3. *TTN* missense variants in subjects meeting criteria for nonischemic dilated cardiomyopathy. Nonischemic dilated cardiomyopathy was defined by left ventricular ejection fraction <50% and left ventricular end diastolic diameter (LVEDD) greater than 2 standard deviations above the sex-specific mean, as well as coronary angiogram confirming the absence of obstructive coronary artery disease.

Supplementary file 4. Parameter estimates for univariable and multivariable Cox proportional

hazard models of atrial fibrillation and heart failure-related hospitalizations. A partially adjusted multivariable model contained covariates of age and sex, and the fully adjusted model additionally accounted for race-ethnicity and ejection fraction <50% closest to AF diagnosis.

Supplementary file 5. Cox proportional hazard models of hospitalizations related to TTN missense variants based on in silico prediction of impact. REVEL score of ≥0.70 indicates potentially deleterious effect. A partially adjusted multivariable model contained covariates of age and sex, and the fully adjusted model additionally accounted for race-ethnicity and ejection fraction <50% closest to AF diagnosis.

Supplementary file 6. Cox proportional hazard models of hospitalizations excluding cases with nonischemic dilated cardiomyopathy. A total of 12 subjects were excluded. A partially adjusted multivariable model contained covariates of age and sex, and the fully adjusted model additionally accounted for race-ethnicity and ejection fraction <50% closest to AF diagnosis.

Supplementary file 7. *TTN*-T32756I variant information.

MDAR checklist

### Data availability

Availability of deidentified individual-level clinical and whole exome sequencing is subject to restrictions from the UIC Institutional Review Board and cannot be publicly released due to privacy and ethical concerns. Clinical data that support the findings of this study are available from the corresponding author upon reasonable request and completion of a data use agreement. However, aggregate data that preserve individual privacy are incorporated inside the manuscript and supplemental materials. The ClinVar accession number for the variant studied is VCV000178164.47. The RNA-Seq data reported in this article were deposited into the NCBI's Gene Expression Omnibus (GEO) database with the accession number GSE312917. Electrophysiology, microscopy, and other raw and analyzed data can be accessed at Mendeley Data (DOI:10.17632/3s8w9kr5hz.1, https://data.mendeley.com/datasets/3s8w9kr5hz/1). Code Availability: R version 4.2.1 and packages (arsenal [*Heinzen et al., 2021a*]; ggplot2 [*Heinzen et al., 2021b*]; ggsurvfit [*Sjoberg et al., 2025a*]; gtsummary [*Sjoberg et al., 2025b*]; reda [*Wang et al., 2025*]; survival [*Therneau et al., 2026*]; tidycmprsk [*Sjoberg and Fei, 2025c*]) are available at https://cran.r-project.org/. Variants are annotated with Ensembl Variant Effect Predictor (https://www.ensembl.org/info/docs/tools/vep/index.html) and CardioDB (https://cardiodb.org/titin) for TTN region and percent spliced in (PSI).

The following datasets were generated:

| Author(s) | Year | Dataset title | Dataset URL | Database and Identifier |
|---|---|---|---|---|
| Arif M | 2025 | A titin missense variant drives atrial electrical remodeling and is associated with atrial fibrillation | https://doi.org/10.17632/3s8w9kr5hz.1 | Mendeley Data, 10.17632/3s8w9kr5hz.1 |
| Pavel AM | 2025 | A Titin Missense Variant Causes Atrial Fibrillation | https://www.ncbi.nlm.nih.gov/geo/query/acc.cgi?acc=GSE312917 | NCBI Gene Expression Omnibus, GSE312917 |
| National Center for Biotechnology Information | 2026 | NM_001267550.2(TTN):c.98267C>T (p.Thr32756Ile) | https://www.ncbi.nlm.nih.gov/clinvar/variation/VCV000178164.47/?redir=vcv | ClinVar, VCV000178164.47 |

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
