## [Editor Report · eLife Assessment]

The study presents **important** findings regarding the incidence and clinical impact of a mutation in a cardiac muscle protein and its association with the development of atrial fibrillation. The authors provide **convincing** evidence of electrophysiological disturbances in cells with this mutation and of its association with atrial fibrillation, which would be of interest to cardiologists. Evidence supporting the conclusion that this mutation causes atrial fibrillation would benefit from more rigorous electrophysiologic approaches.

---

## [Referee Report · Reviewer #1 (Public review)]

Pavel et al. analyzed a cohort of atrial fibrillation (AF) patients from the University of Illinois at Chicago, identifying TTN truncating variants (TTNtvs) and TTN missense variants (TTNmvs). They reported a rare TTN missense variant (T32756I) associated with adverse clinical outcomes in AF patients. To investigate its functional significance, the authors modeled the TTN-T32756I variant using human induced pluripotent stem cell-derived atrial cardiomyocytes (iPSC-aCMs). They demonstrated that mutant cells exhibit aberrant contractility, increased activity of the cardiac potassium channel KCNQ1 (Kv7.1), and dysregulated calcium homeostasis. Interestingly, these effects occurred without compromising sarcomeric integrity. The study further identified increased binding of the titin-binding protein Four-and-a-Half Lim domains 2 (FHL2) with KCNQ1 and its modulatory subunit KCNE1 in the TTN-T32756I iPSC-aCMs.

Comments on revised version:

This revised manuscript demonstrates significant improvement, notably through the inclusion of new data (Supplementary Figures 5 and 7) and expanded explanations in the main text. These additions strengthen the association between the TTN-T32756I missense variant and electrophysiological phenotypes relevant to atrial fibrillation (AF). The authors are commended for their thorough and thoughtful responses to reviewer feedback, their transparency in acknowledging limitations, and their efforts to provide mechanistic insight into the observed phenotype.

Nonetheless, several important limitations remain and should be more explicitly addressed when framing the conclusions and selecting the final manuscript title:

(1) While the data support a functional impact of the TTN-T32756I variant, the evidence does not yet definitively establish causality in the context of AF. Statements asserting a causal relationship should be softened and clearly framed as suggestive, pending further in vivo or patient-specific validation.

(2) The study models the TTN-T32756I variant in a single healthy iPSC line using CRISPR/Cas9 editing. Although this provides a genetically controlled system, the absence of validation in patient-derived iPSCs or replication across multiple isogenic lines limits the generalizability and reproducibility of the findings.

(3) The co-localization and co-immunoprecipitation (co-IP) data provide strong support for an interaction between FHL2 and the KCNQ1/KCNE1 complex. However, in the current form, the proposed mechanism remains plausible but not fully validated.

---

## [Referee Report · Reviewer #2 (Public review)]

Summary:

The authors present data from a single-center cohort of African-American and Hispanic/Latinx individuals with atrial fibrillation (AF). This study provides insight into the incidences and clinical impact of missense variants in the Titin (TTN) gene in this population. In addition, the authors identified a single amino acid TTN missense variant (TTN-T32756I) that was further studied using human induced pluripotent stem cell-derived atrial cardiomyocytes (iPSC-aCMs). These studies demonstrated that the Four-and-a-Half Lim domains 2 (FHL2), has increased binding with KCNQ1 and its modulatory subunit KCNE1 in the TTN-T32756I-iPSC-aCMs, enhancing the slow delayed rectifier potassium current (Iks) and is a potential mechanism for atrial fibrillation. Finally, the authors demonstrate that suppression of FHL2 could normalize the Iks current.

Strengths:

The strengths of this manuscript/study are listed below:

(1) This study includes a previously underrepresented population in the study the genetic and mechanistic basis of AF.

(2) The authors utilize current state-of-the-art methods to investigate the pathogenicity of a specific TTN missense variant identified in this underrepresented patient population.

(3) The findings of this study identify a potential therapeutic for treating atrial fibrillation.

Weaknesses:

(1) The authors do not include a non-AF group when evaluating the incidence and clinical significance of TTN missense variants in AF patients. The authors appropriately acknowledge this as a limitation in their single-center cohort.

(2) All other concerns from a previous version of this manuscript have been adequately addressed by the authors in this revision.

---

## [Author Response]

The following is the authors’ response to the original reviews.

**Reviewer #1 (Public review):**
Summary:Pavel et al. analyzed a cohort of atrial fibrillation (AF) patients from the University of Illinois at Chicago, identifying TTN truncating variants (TTNtvs) and TTN missense variants (TTNmvs). They reported a rare TTN missense variant (T32756I) associated with adverse clinical outcomes in AF patients. To investigate its functional significance, the authors modeled the TTN-T32756I variant using human induced pluripotent stem cell-derived atrial cardiomyocytes (iPSC-aCMs). They demonstrated that mutant cells exhibit aberrant contractility, increased activity of the cardiac potassium channel KCNQ1 (Kv7.1), and dysregulated calcium homeostasis. Interestingly, these effects occurred without compromising sarcomeric integrity. The study further identified increased binding of the titin-binding protein Four-and-a-Half Lim domains 2 (FHL2) with KCNQ1 and its modulatory subunit KCNE1 in the TTN-T32756I iPSCaCMs.Strengths:This work has translational potential, suggesting that targeting KCNQ1 or FHL2 could represent a novel therapeutic strategy for improving cardiac function. The findings may also have broader implications for treating patients with rare, disease-causing variants in sarcomeric proteins and underscore the importance of integrating genomic analysis with experimental evidence to advance AF research and precision medicine.Weaknesses(1) Variant Identification: It is unclear how the TTN missense variant (T32756I) was identified using REVEL, as none of the patients' parents reportedly carried the mutation or exhibited AF symptoms. Are there other TTN variants identified in the three patients carrying TTN-T32756I? Clarification on this point is necessary.

We thank the reviewer for their insightful comment. We have now clarified these in the method section.

Line 484-491: “The TTN-T32756I variant (REVEL Score: 0.58758, Supplementary Table 1) was prioritized due to its occurrence in multiple unrelated individuals within our clinical AF cohort, despite no reported family history of AF in affected individuals. While no parental inheritance was observed, the possibility of de novo origin cannot be excluded. Furthermore, this variant is located within a region overlapping a deletion mutation recently shown to cause AF in a zebrafish model, supporting its potential pathogenicity [37]. Notably, the affected individuals did not carry additional loss-of-function TTN variants.”

(2) Patient-Specific iPSC Lines: Since the TTN-T32756I variant was modeled using only one healthy iPSC line, it is unclear whether patient-specific iPSC-derived atrial cardiomyocytes would exhibit similar AF-related phenotypes. This limitation should be addressed.

We have now acknowledged this limitation in the revised manuscript.

Line 505-509: “Due to the patients' unavailability of peripheral blood mononuclear cells (PBMCs), we utilized a healthy iPSC line and introduced the TTN-T32756I variant using CRISPR/Cas9 genome editing. This approach ensures an isogenic background, thereby minimizing genetic variability and providing a controlled system to study the direct effects of the mutation.”

(3) Hypertension as a Confounding Factor: The three patients carrying TTN-T32756I also have hypertension. Could the hypertension associated with this variant contribute secondarily to AF? The authors should discuss or rule out this possibility.

We have now explicitly discussed this in the revised manuscript.

Line 362-367: “Hypertension is a common comorbidity in patients with AF and could contribute to disease progression. However, all three individuals carrying TTN-T32756I exhibited earlyonset AF (onset before 66 years), with one case occurring as early as 36 years. This suggests a potential two-hit mechanism, where genetic predisposition and comorbidities influence disease risk. Importantly, our iPSC model isolates the genetic effects of TTN-T32756I from other factors, supporting a direct pathogenic role.”

(4) FHL2 and KCNQ1-KCNE1 Interaction: Immunostaining data demonstrating the colocalization of FHL2 with the KCNQ1-KCNE1 (MinK) complex in TTN-T32756I iPSC-aCMs are needed to strengthen the mechanistic findings.

We thank the reviewer for this insightful suggestion. We agree that additional immunostaining data would further strengthen the evidence for FHL2 colocalization with the KCNQ1-KCNE1 complex in TTN-T32756I iPSC-aCMs. In line with this, we have expanded our analysis to include both co-immunoprecipitation and confocal microscopy. As described in the revised manuscript (Lines 282–287), the colocalization between KCNE1 and FHL2 was increased by approximately threefold in TTN-T32756I iPSC-aCMs compared with WT, supporting an enhanced interaction between these proteins (Figure 5A, Supplementary Figure 6). We are generating additional immunostaining data to validate and extend these findings, and we will incorporate them into the revised submission to further substantiate the mechanistic link proposed.

Line 282-287: “…..if TTN-T32756I increases I_ks_ by modulating the interaction between KCNQ1KCNE1 and FHL2, we performed co-immunoprecipitation studies and confocal microscopy in both WT and TTN-T32756I-iPSC-aCMs. The co-localization between KCNE1 and FHL2 increased ~3 fold in TTN-T32756I-iPSC-aCMs, suggesting an increased interaction between them (Figure 5A, Supplementary Figure 7).”

(5) Functional Characterization of FHL2-KCNQ1-KCNE1 Interaction: To further validate the proposed mechanism, additional functional assays are necessary to characterize the interaction between FHL2 and the KCNQ1-KCNE1 complex in TTN-T32756I iPSC-aCMs.

We thank the reviewer for this valuable suggestion. We agree that additional functional assays would provide further validation of the proposed mechanism. However, we believe such in-depth characterization warrants a dedicated follow-up study and is beyond the scope of the current revision. In this work, our primary objective is to establish that the TTN missense variant can exert a detrimental effect and serve as a substrate for AF.

Line 418-419: “Further study is needed to validate the proposed mechanism and determine if TTNmvs in other regions are associated with AF by a similar process.”

**Reviewer #2 (Public review):**
Summary:The authors present data from a single-center cohort of African-American and Hispanic/Latinx individuals with atrial fibrillation (AF). This study provides insight into the incidences and clinical impact of missense variants in this population in the Titin (TTN) gene. In addition, the authors identified a single amino acid TTN missense variant (TTN-T32756I) that was further studied using human induced pluripotent stem cell-derived atrial cardiomyocytes (iPSC-aCMs). These studies demonstrated that the Four-and-a-Half Lim domains 2 (FHL2) has increased binding with KCNQ1 and its modulatory subunit KCNE1 in the TTN-T32756I-iPSCaCMs, enhancing the slow delayed rectifier potassium current (Iks) and is a potential mechanism for atrial fibrillation. Finally, the authors demonstrate that suppression of FHL2 could normalize the Iks current.Strengths:The strengths of this manuscript/study are listed below:(1) This study includes a previously underrepresented population in the study of the genetic and mechanistic basis of AF.(2) The authors utilize current state-of-the-art methods to investigate the pathogenicity of a specific TTN missense variant identified in this underrepresented patient population.(3) The findings of this study identify a potential therapeutic for treating atrial fibrillation.Weaknesses:(1) The authors do not include a non-AF group when evaluating the incidence and clinical significance of TTN missense variants in AF patients.

We appreciate the reviewer’s comment and acknowledge the limitation of not including a non-AF control group in our clinical analysis. As noted in the revised manuscript (Lines 347–353), our cohort was derived from a single-center registry of individuals with AF and therefore lacks a matched non-AF control population for direct comparison of TTN missense variant incidence. We agree that future studies incorporating larger, multiethnic validation cohorts with both AF and non-AF individuals, as well as evaluating AF-specific measures such as arrhythmia burden and treatment response, will be essential to fully elucidate the clinical significance of TTN missense variants in AF.

Line 347-353: “Our cohort is derived from a single-center multi-ethnic registry of individuals with AF and lacks a matched cohort of non-AF controls to compare the incidence of TTN missense variants. Further study exploring these associations in mult-ethnic, larger validation cohorts that include both AF and non-AF individuals and examining AF-specific measures such as arrhythmia burden or treatment response will be necessary to fully understand the clinical importance of TTNmvs in AF.”

(2) The authors do not provide evidence that TTN-T32756I-iPSC-aCMs are arrhythmogenic, only that there is an increase in the Iks current and associated action potential changes. More specifically, the authors report that "compared to the WT, TTN-T32756I-iPSC-aCMs exhibited increased arrhythmic frequency," yet it is unclear what they are referring to by "arrhythmic frequency."

We thank the reviewer for this important point and for highlighting the need for clarification. In our study, the term “arrhythmic frequency” was intended to describe the increased spontaneous beating rate, irregular action potential patterns, and abnormal calcium handling observed in TTN-T32756I iPSC-aCMs compared with WT. These findings support the concept that the AF-associated TTN-T32756I variant promotes ion channel remodeling and perturbs excitation–contraction coupling, thereby creating a potential arrhythmogenic substrate for AF. To avoid ambiguity, we have removed the term “arrhythmic frequency” and revised the text for clarity and precision (Lines 222–223).

Lines 222-223: “Compared to the WT, TTN-T32756I-iPSC-aCMs exhibited increased frequency along with a significant reduction of the time to 50% and 90% decline of calcium transients (Figure 3G-I, Supplementary Figure 4F).”

(3) There seem to be discrepancies regarding the impact of the TTN-T32756I variant on mechanical function. Specifically, the authors report "both reduced contraction and abnormal relaxation in TTN-T32756I-iPSC-aCMs" yet, separately report "the contraction amplitude of the mutant was also increased . . . suggesting an increased contractile force by the TTN-T32756IiPSC-aCMs and TTN-T32756I-iPSC-CMs exhibited similar calcium transient amplitudes as the WT."

We thank the reviewer for highlighting this critical point and apologize for the lack of clarity. We intended to distinguish between changes in contractile force and contractile dynamics. Specifically, the increased contraction amplitude observed in TTN-T32756I iPSCaCMs reflects enhanced contractile force, whereas the reduced contraction duration and impaired relaxation reflect abnormalities in contractile kinetics. Together, these findings indicate that the TTN-T32756I variant alters both the strength and the temporal dynamics of contraction, consistent with dysfunctional mechanical performance. We have revised the text accordingly to more accurately convey these results (Lines 187–192).

Lines 187-192: “Compared to WT, the beating frequency of the TTN-T32756I-iPSC-aCMs was significantly increased (52 ± 7.8 vs. 98 ± 7.5 beats per min, P=0.001; Figure 2C) coupled with the reduction of the contraction duration (456.5 ± 61.45 vs 262.9 ± 48.16 msec, P=0.032; Figure 2D), the peak-to-peak time (1529 ± 195.5 vs 636.6 ± 135.8 msec, P=0.004; Supplementary Figure 3B), and the relaxation (281.5 ± 42.95 vs 79.40 ± 21.14 msec, P=0.003; Supplementary Figure 3A).”

**Reviewer #3 (Public review):**
Summary:The authors describe the abnormal contractile function and cellular electrophysiology in an iPSC model of atrial myocytes with a titin missense variant. They provide contractility data by sarcomere length imaging, calcium imaging, and voltage clamp of the repolarizing current iKs. While each of the findings is interesting, the paper comes across as too descriptive because there is no data merging to support a cohesive mechanistic story/statement, especially from the electrophysiological standpoint. There is not enough support for the title "A Titin Missense Variant Causes Atrial Fibrillation", since there is no strong causative evidence. There is some interesting clinical data regarding the variant of interest and its association with HF hospitalization, which may lead to future important discoveries regarding atrial fibrillation.Strengths:The manuscript is well written, and a wide range of experimental techniques are used to probe this atrial fibrillation model.Weaknesses(1) While the clinical data is interesting, it is essential to rule out heart failure with preserved EF as a confounder. HFpEF leads to AF due to increased atrial remodeling, so the fact that patients with this missense variant have increased HF hospitalizations does not necessarily directly support the variant as causative of AF. It could be that the variant is associated directly with HFpEF instead, and this needs to be addressed and corrected in the analyses.

We appreciate the reviewer’s insightful comment and agree that HFpEF-related atrial remodeling could represent a potential confounder in the association between TTN missense variants and AF. The primary aim of our clinical analysis was to assess the potential significance of TTNmv in AF, recognizing the inherent limitations of retrospective observational data in establishing causality. To complement this, our in vitro studies were specifically designed to demonstrate that TTNmv can alter the electrophysiological substrate, thereby predisposing to AF independent of clinical comorbidities.

While HFpEF is an important consideration, to our knowledge, no existing literature directly implicates TTNmv in HFpEF pathogenesis. In contrast, loss-of-function TTN variants are more commonly associated with HFrEF and dilated cardiomyopathy, and even these associations remain an area of active debate. To address potential confounding in our cohort, we adjusted for reduced ejection fraction in multivariable analyses of clinical outcomes. Additionally, we performed a sensitivity analysis excluding patients with nonischemic dilated cardiomyopathy (Supplementary Table 6). Together, these approaches mitigate the potential impact of heart failure subtypes on our findings, while our mechanistic studies strengthen the argument that TTNmv may contribute directly to AF susceptibility.

(2) All contractility and electrophysiologic data should be done with pacing at the same rate in both control and missense variant groups, to control for the effect of cycle length on APD and calcium loading. A shorter APD cannot be claimed when the firing rate of one set of cells is much faster than the other, since shorter APD is to be expected with a quicker rate. Similarly, contractility is affected by diastolic interval because of the influence of SR calcium content on the myocyte power stroke. So the cells need to be paced at the same rate in the IonOptix for any direct comparison of contractility. The authors should familiarize themselves with the concept of electrical restitution.

We thank the reviewer for this crucial technical comment. iPSC-derived cardiomyocytes (iPSC-CMs) are known to exhibit spontaneous automaticity due to the presence of pacemaker-like currents and reduced I_K1_, which enables interrogation of their intrinsic electrophysiological properties and disease-relevant remodeling. In our study, we leveraged this feature to test the hypothesis that TTN missense variants alter electrophysiological properties through ion channel remodeling. That said, we fully agree with the reviewer that pacing iPSCCMs at a controlled cycle length is essential for minimizing rate-dependent effects on APD, calcium handling, and contractility, and would improve the interpretability of group comparisons. While iPSC-CMs with matched genetic backgrounds are expected to display broadly comparable electrophysiological profiles, biological and technical variability can influence spontaneous beating rates, thereby confounding direct comparisons. To address this, we have incorporated pacing protocols into our revised experimental design to ensure that APD and contractility measurements are obtained under identical cycle lengths, consistent with the concept of electrical restitution.

(3) It is interesting that the firing rate of the myocytes is faster with the missense variant. This should lead to a hypothesis and investigation of abnormal automaticity or triggered activity, which may also explain the increased contractility since all these mechanisms are related to the SR's calcium clock and calcium loading. See #2 above for suggestions on how to probe calcium handling adequately. Such an investigation into impulse initiation mechanisms would be compelling in supporting the primary statement of the paper since these are actual mechanisms thought to cause AF.

We thank the reviewer for this insightful suggestion. We agree that the faster firing rate observed in TTN-T32756I iPSC-aCMs raises the possibility of abnormal automaticity or triggered activity, both of which are highly relevant to AF pathophysiology. As these mechanisms are tightly coupled to calcium handling and the SR calcium clock, further probing of calcium cycling abnormalities would provide valuable mechanistic insights. While this level of investigation is beyond the scope of the current study, we view it as a compelling future direction that could directly link TTN missense variants to impulse initiation abnormalities contributing to AF.

(4) The claim of shortened APD without correcting for cycle length is problematic. However, linking shortened APD in isolated cells alone to AF causation is more complicated. To have a setup for reentry, there must be a gradient of APD from short to long, and this can only be demonstrated at the tissue level, not at the cellular level, so reentry should not be invoked here. If shortened APD is demonstrated with correction of the cycle length problem, restitution curves can be made showing APD shortening at different cycle lengths. If restitution is abnormal (i.e. the APD does not shorten normally in relation to the diastolic interval), this may lead to triggered activity which is an arrhythmogenic mechanism. This would also tie in well with the finding of abnormally elevated iKs current since iKs is a repolarizing current directly responsible for restitution.

We thank the reviewer for this necessary clarification. We agree that isolated cell studies cannot directly demonstrate reentrant circuits and that reentry should not be inferred solely from cellular APD data. Our observation of shortened APD and abnormal beating patterns in TTN-T32756I iPSC-aCMs suggests ion channel remodeling that may predispose to arrhythmogenic conditions. Still, we recognize that tissue-level gradients of APD are required to establish reentry as a mechanism. Accordingly, we have removed mention of “the reentrant mechanism” from the revised manuscript and limited our interpretation to the cellular findings. Future studies incorporating pacing protocols and restitution curve analyses will be valuable in determining whether abnormal APD restitution and elevated I_Ks_ contribute to triggered activity, thereby providing a more direct mechanistic link to AF (Lines 101–105).

Lines 101-105: “Our study showed that the TTN-T32756I iPSC-aCMs exhibited a striking AF-like EP phenotype in vitro, and transcriptomic analyses revealed that the TTNmv increases the activity of the FHL2, which then modulates the slow delayed rectifier potassium current (I_Ks_) to cause AF.”

**Reviewer #1 (Recommendations for the authors)**:Electrophysiological Phenotype in Ventricular CMs: Has the iPSC line carrying TTN-T32756I been differentiated into ventricular cardiomyocytes (iPSC-vCMs)? The reported cellular phenotype in iPSC-aCMs does not seem to specifically reflect an AF phenotype. Does the variant produce similar electrophysiological alterations in iPSC-vCMs?

We thank the reviewer for this thoughtful comment. To date, we have not differentiated the TTN-T32756I iPSC line into ventricular cardiomyocytes (iPSC-vCMs). Our current work focuses on iPSC-aCMs, where we demonstrate that the AF-associated TTNT32756I variant induces ion channel remodeling and abnormal beating patterns, thereby creating a potential arrhythmogenic substrate relevant to AF. We agree that investigating whether this variant produces similar or distinct electrophysiological alterations in iPSC-vCMs would provide essential insights into chamber-specific effects and broaden our mechanistic understanding. We have acknowledged this as a future direction in the revised manuscript (Lines 422–425).

Lines 422-425: “While we have not yet explored the effect of TTN-T32756I in iPSC-derived ventricular cardiomyocytes, it would be interesting to investigate whether this variant produces similar or distinct electrophysiological alterations in the ventricular cardiomyocytes.”